# ^18^F-Radiolabeled Translocator Protein (TSPO) PET Tracers: Recent Development of TSPO Radioligands and Their Application to PET Study

**DOI:** 10.3390/pharmaceutics14112545

**Published:** 2022-11-21

**Authors:** Truong Giang Luu, Hee-Kwon Kim

**Affiliations:** 1Department of Nuclear Medicine, Molecular Imaging & Therapeutic Medicine Research Center, Jeonbuk National University Medical School and Hospital, Jeonju 54907, Republic of Korea; 2Research Institute of Clinical Medicine of Jeonbuk National University-Biomedical Research Institute of Jeonbuk National University Hospital, Jeonju 54907, Republic of Korea

**Keywords:** translocator protein, TSPO ligands, PET, neuroinflammation, molecular imaging

## Abstract

Translocator protein 18 kDa (TSPO) is a transmembrane protein in the mitochondrial membrane, which has been identified as a peripheral benzodiazepine receptor. TSPO is generally present at high concentrations in steroid-producing cells and plays an important role in steroid synthesis, apoptosis, and cell proliferation. In the central nervous system, TSPO expression is relatively modest under normal physiological circumstances. However, some pathological disorders can lead to changes in TSPO expression. Overexpression of TSPO is associated with several diseases, such as neurodegenerative diseases, neuroinflammation, brain injury, and cancers. TSPO has therefore become an effective biomarker of related diseases. Positron emission tomography (PET), a non-invasive molecular imaging technique used for the clinical diagnosis of numerous diseases, can detect diseases related to TSPO expression. Several radiolabeled TSPO ligands have been developed for PET. In this review, we describe recent advances in the development of TSPO ligands, and ^18^F-radiolabeled TSPO in particular, as PET tracers. This review covers pharmacokinetic studies, preclinical and clinical trials of ^18^F-labeled TSPO PET ligands, and the synthesis of TSPO ligands.

## 1. Introduction

Mitochondrial membrane proteins play a crucial role in maintaining mitochondrial homeostasis, and errors in these proteins can lead to various diseases. Mitochondrial membrane proteins have therefore become the subject of intense interest in the study of disease diagnosis and treatment [1]. The peripheral benzodiazepine receptor (PBR), a mitochondrial transmembrane protein first described in 1977 [2], plays a crucial role in multiple complex physiological processes, including cholesterol translocation from the cytoplasm to mitochondria and the synthesis of neurosteroids [3]. The protein is now known as a translocator protein (TSPO).

TSPO comprises 169 amino acids and 5 transmembrane (TM) α-helices with a pocket accepting a ligand in the middle [4]. The five α-helices are linked by extramitochondrial and intramitochondrial loops with the N terminal at TM1 and the C-terminal at TM5. TSPO can be found in contact positions between the outer and inner mitochondrial membranes of steroidogenic tissues.

TSPO functions as a mitochondrial membrane transport channel for cholesterol [5], but it plays a role in several other physiological functions, including immunomodulation [6], cell proliferation and differentiation [3], apoptosis [7], protein import, and ion transport [8]. TSPO can be found all over the body, with large quantities in steroidogenic tissue, and it is often overexpressed in the kidneys, adrenal glands, lungs, and heart. However, recent studies evaluating TSPO function through genetics have raised questions about the true role of TSPO in steroidogenesis, as well as several other functions [9,10,11]. Therefore, these TSPO functions need to be carefully re-evaluated.

TSPO overexpression is reportedly associated with diseases such as brain ischemia damage [12]. In addition, TSPO is related to various diseases of the central nervous system (CNS) [13,14]. TSPO overexpression is found in activated microglia, and upregulation of TPSO has been observed in astrocytes. Because microglia activation is related to various CNS-related diseases, TSPO overexpression in activated microglia and astrocytes can be used as an indicator of neuroinflammation and neurodegenerative disorders, such as Alzheimer’s disease (AD), Parkinson’s disease (PD), multiple sclerosis (MS), Huntington’s disease (HD), and others [15]. TSPO overexpression is also associated with several tumors, including breast and colon cancers, and it is considered a cancer biomarker [16].

Positron emission tomography (PET) is a powerful tool for monitoring biochemical phenomena [17,18,19,20]. It is a non-invasive medical imaging technique, which uses radioactive isotopes of pharmaceuticals (tracers) that provide quantitative information on biological processes in the form of high-resolution, real-time imagery [21,22,23]. PET is widely used in the diagnosis of cancers, brain and cardiovascular diseases, the evaluation of therapy, and the development of medication [24].

Several studies have demonstrated that PET imaging can be useful in studying CNS-related diseases [25,26,27], and a variety of TSPO ligands have been developed [15]. TSPO ligands bearing ^11^C and ^18^F radioisotopes are now widely employed in PET. Although several TSPO ligands bearing the ^11^C radioisotope, which meets the requirements of PET imaging, are used clinically, they suffer from intrinsic limitations, such as a short half-life (t_1/2_) of 20.4 min, which prevents wide-scale utilization in PET.

TSPO ligands with the ^18^F radioisotope have therefore received significant research attention. ^18^F offers advantages over ^11^C [24,28], including a longer half-life (t_1/2_ = 109.7 min), making extended dynamic PET studies possible, and a lower positron energy (650 keV). In addition, ^18^F can be stored for longer periods and can be sent to relatively distant facilities. A variety of TSPO ligands radiolabeled by ^18^F have been developed and used for PET imaging in the diagnosis of multiple diseases. In this review, we describe recent advances in developing TSPO ligands bearing ^18^F.

## 2. TSPO as an Indicator for Neuroinflammation

Neuroinflammation is a CNS defense mechanism, which is triggered by pathogens, such as toxic metabolites, infection, and traumatic brain injury [29]. The blood–brain barrier, a specialized tissue made of astrocytes and endothelial cells, is normally considered an armor that shields the CNS. If the barrier is breached, potentially dangerous agents can enter the brain’s delicate environment [30]. In that event, innate immune mechanisms will be activated to respond to these agents to enhance the expression of microglia cells and cause inflammation. Neuroinflammation is a common symptom of diseases of the CNS, such as AD, PD, and HD [31].

Microglia are neuroglia that represent 10–15% of all cells in the brain [32]. As resident macrophage cells, they are the first and most important line of immune defense in the CNS. Their primary role is to detect factors that can harm the CNS. When microglia change from a resting state to an active phenotype, they exhibit a significant shift in morphology. Microglia move to the damaged location and perform phagocytic functions, “eating” plaque, damaged or superfluous neurons, synapses, and infectious pathogens [33]. The activation of microglial cells in the CNS, particularly at inflammatory sites, is considered a biomarker of neuroinflammation.

Although previous studies showed that high TSPO expression is closely related to the activation of microglia [33], and many studies suggested that increased expression of TSPO represents an activation of microglial cells or increased neuroinflammation [34], a few recent studies showed that microglia activation is not necessarily associated with overexpression of TSPO in individual microglial cells [35]. However, TSPO is still considered a useful biomarker of neuroinflammation and related diseases.

Assessment of activated microglia is important for the treatment of various disorders associated with microglia. PET imaging using TSPO is effective because PET can provide visual and numerical data on biological events and disease progression related to neuroinflammation. PET using TSPO ligands has therefore been widely used to detect neuroinflammation and related diseases.

## 3. Development of ^18^F-Radiolabeled TSPO Ligands

Various TSPO ligands with radioisotopes, which can bind strongly to TSPO, have been developed for use in PET imaging studies. Among the developed TSPO ligands, PK-11195 (1-(2-chlorophenyl)-N-methyl-N-(1-methylpropyl)-3-isoquinoline carboxamide) and Ro5-4864 (4′-chlorodiazepam) are considered first-generation ligands (Figure 1). These first-generation ligands were radiolabeled with ^11^C and met the initial needs of PET imaging studies.

PK11195 is a carboxamide isoquinoline, which was first reported in 1984 [36]. Many studies have found that PK11195 demonstrates high affinity and selectivity to TSPO for PET study [37]. Because the binding affinity of Ro5-4864 varies by temperature and species [38,39], PK11195 is the best studied ligand, and it has been used as a reference to validate other TSPO ligands in diagnostic studies of CNS-related diseases [40]. However, some disadvantages of PK11195, such as low brain uptake, poor penetration of the blood–brain barrier, a short half-life, and highly variable kinetic behavior, have prevented wide-scale utilization of this ligand [12] and encouraged scientists to develop the alternative, second-generation TSPO ligands (Table 1).

**Figure 1 pharmaceutics-14-02545-f001:**
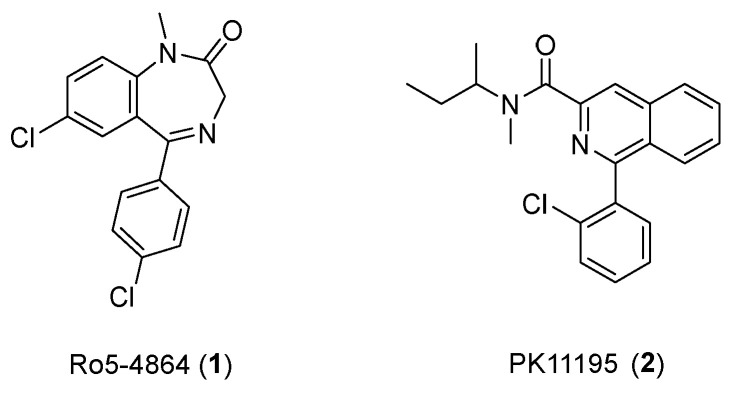
Structures of Ro5-4864 and PK11195.

**Table 1 pharmaceutics-14-02545-t001:** Summary of ^18^F-radiolabeled TSPO ligands.

Chemical Class	TSPO Ligand	Binding Affinity/Lipophilicity	Stage of Research (Preclinical/Clinical)	Comments	Ref.
Phenoxyarylacetamides	[^18^F]FEPPA	K_i_ = 0.07 nM Log P = 2.99	LPS mouse PD patients AD patients Psychosis patients	High binding affinity Suitable lipophilicity for brain penetration Effective in clinical application	[41,42,43,44,45]
	[^18^F]Fluoromethyl-PBR28	K_i_ = 1.85 nM Log D = 2.85	LPS mouse (EAM) rats Ischemic stroke rat	Suitable lipophilicity Effective in clinical application Lack of clinical studies	[46,47]
	[^18^F]FEMPA	-	Atherosclerotic plaques in mice AD patients Friedreich ataxia patients	Rapid blood clearance and uptake High binding sensitivity to the human gene polymorphism rs6971 Effective in clinical application	[48,49,50]
	[^18^F]FEDAA1106	K_i_ = 0.078 nM Log D = 3.81	Rats MS patients AD patients	High binding affinity High lipophilicity Ineffective in clinical application	[51,52,53,54,55]
	[^18^F]DAA1106	K_i_ = 0.043 nM Log P = 3.65	Ischemic rats	High lipophilicity Lack of clinical studies	[56,57,58]
	[^18^F]PBR06	K_i_ = 0.997 nM Log D = 4.05	Stroke mouse HD mice MCAO mice AD mice Monkey MS patients	High lipophilicity Effective in preclinical and clinical studies	[59,60,61,62,63,64,65]
Pyrazolopyrimidines	[^18^F]DPA-714	Ki = 7.0 nM Log D = 2.44	Mice, Monkey, Human PACNS patients AD patients Stroke patients MS patients PD patients ALS patients	Suitable lipophilicity Rapid penetration and good retention in the brain Effective in monitoring and diagnosis for many neurological diseases	[66,67,68,69,70,71,72,73,74,75,76,77]
	[^18^F]VUIIS-1008	Ki = 0.27 nM Log D = 2.5	C6 Glioma-bearing rats	Suitable binding affinity and lipophilicity Effective in preclinical studies Lack of clinical studies	[78,79]
	[^18^F]DPA-C5yne	Ki = 0.35 nM Log P = 2.39	Rat	Suitable binding affinity and lipophilicity Effective in preclinical studies Lack of clinical studies	[80,81]
	[^18^F]F-DPA	Ki = 1.7 nM Log D = 2.34	Sprague Dawley Rat Neuropathic pain-induced rats Cerebral ischemia mice AD mice	Suitable lipophilicity Effective in preclinical studies Lack of clinical studies	[82,83,84,85,86]
Imidazopyridine acetamides	[^18^F]PBR102	Ki = 5.8 ± 0.4 Log P = 2.7 ± 0.1	Rat Excitotoxin neuroinflammation mice Non-human primates Human	Suitable lipophilicity Effective in preclinical studies Good preclinical effect for many species	[87,88,89]
	[^18^F]PBR111	Ki= 3.2 ± 0.4 nM Log P = 3.2 ± 0.1	Rat, Ops rat Non-human primates Human Schizophrenia patients Psychosis patients MS patients	Suitable lipophilicity Effective in preclinical studies Good preclinical effect for many species Effective in clinical application	[87,88,89,90,91,92]
	[^18^F]PBR316	Ki = 6.0 ± 1.4 nM Log P = 2.16 ± 0.07	Rats	Lack of preclinical and clinical studies	[93]
	[^18^F]CB251	Ki = 0.27 ± 0.09 nM Log D = 3.00 ± 0.03	Neuroinflammation rats Human glioblastoma	High binding affinity	[94,95,96]
	[^18^F]BS224	Ki = 0.51 ± 0.03 nM Log D = 2.78 ± 0.04	LPS rats Ischemic stroke rats	Suitable binding affinity and lipophilicity Effective in preclinical studies Lack of clinical studies	[97]
Oxopurine	[^18^F]FEDAC	Ki = 1.3 nM Log D = 3.2	Collagen arthritis mice Neuroinflammatory rat Monkey Atherosclerosis rabbit Human liver cell Acute myocardial infarction patients	Effective in preclinical studies with many species Lack of clinical studies	[98,99,100,101]
Acetamidobenzoxazolone	[^18^F]FEBMP	Ki = 6.6 ± 0.7 nM Log D = 3.4	Ischemic rats MCAO rats AD mice	Effective in preclinical studies Lack of clinical studies	[102,103,104,105]
	[^18^F]FPBMP	Ki = 16.7 ± 2.5 nM Log D = 3.5	Ischemic rats	Lack of preclinical and clinical studies	[102,103,104,105]
Pyridazinoindoles	[^18^F]SSR180575	Ki = 1.19 ± 0.05 nM	Rat	Lack of preclinical and clinical studies	[106,107]
Tricyclic indoles	[^18^F]GE180	Ki = 2.4 nM Log D = 2.95	LPS-injected mouse MCAO rats AD mice Pigs Human High-grade glioma patient	Effective in preclinical studies with many species Poor brain penetration in clinical study Clinically effective for some given diseases	[108,109,110,111,112,113,114]
	[^18^F]GE387	Ki = 47.3 ± 7.0 nM	LPS rats Monkeys Humans	Low binding affinity Low binding sensitivity to the human gene polymorphism rs6971	[115,116]
Quinoline carboxamide	[^18^F]AB5186	Ki = 2.8 ± 0.8 nM	Rats Glioma mice Baboon	Effective in preclinical studies Lack of clinical studies	[117,118,119,120]
Isoquinoline carboxamide	(R)- [^18^F] NEBIQUINIDE	Ki = 5.3 ± 0.6 nM Log P = 2.35 ± 0.14	Rats	Low binding sensitivity to the human gene polymorphism rs6971 Lack of preclinical and clinical studies	[121]
Quinazoline carboxamide	[^18^F]ER176	Ki = 3,10 ± 0,30 nM Log D = 3.55 ± 0.02	Rats	High lipophilicity Lack of preclinical and clinical studies	[122,123]

### 3.1. Phenoxyarylacetamides

In 2008, Pike and co-workers reported on [^11^C]PBR28 (N-(2-[^11^C]methoxybenzyl)-N-(4-phenoxypyridin-3-yl)acetamide), a potential second-generation TSPO radioligand based on a phenoxyarylacetamide structure. Basically, the ring opening of the diazepine ring of Ro5-4864 will form high-affinity PBR ligands, including DAA1106. Then, with a pyridine ring replacing one of the benzene rings of DAA1106, the derivatives of phenoxyarylacetamide PBR are generated. This method produced a series of compounds with reduced lipophilicity but still retaining properties such as high affinity for PBR and blood–brain barrier penetration ability [124]. Subsequent studies developed new ^18^F-radiolabeled ligands by replacing the ^11^C on PBR28 with ^18^F (Figure 2).

#### 3.1.1. [^18^F]FEPPA

[^18^F]FEPPA (**3**) was synthesized by Wilson and co-workers in 2008. The initial investigations showed that it has a K_i_ value of 0.07 nM for TSPO, a log P of 2.99 at a pH of 7.4, and a standard uptake value (SUV) of 0.6 at 5 min, suggesting that it is suitable for use as a PET tracer [41].

In 2018, Hosten and co-workers described an improved and automated method of synthesizing [^18^F]FEPPA using the AllInOne module. After 55 min from end of bombardment (EOB), [^18^F]FEPPA was obtained with 34% ± 2% radiochemical yield (non-decay-corrected yield, *n* = 6) [42]. The prepared compound had a molar activity of 198 ± 125 GBq/mol. Its radiochemical purity was >99% after completion of the synthesis process, and the purity remained > 98% in a saline solution for 6 h. Biodistribution and metabolism studies using mice showed that [^18^F]FEPPA was quickly absorbed and excreted by the heart, lungs, and kidney but poorly absorbed by the brain. However, brain absorption in an LPS-injected mouse group was substantially higher (2.2-fold) compared with that in the control animals, and the metabolism of [^18^F]FEPPA in the brain was 4% to 23%, respectively, at 15 and 120 min after injection, which was less than in the plasma. This was a positive sign for PET brain imaging studies of TSPO.

A series of clinical studies have been performed to evaluate the actual effectiveness of [^18^F]FEPPA in various neurological conditions, including PD, AD, and first-episode psychosis [41,42,124]. The studies agreed that the total distribution volume (V_t_) of [^18^F]FEPPA in every brain area was significantly affected by the genotype of the rs6791 polymorphism. The V_t_ value of high-affinity binders (HABs) was higher than that of mixed-affinity binders (MABs) in most brain regions. However, the effectiveness of [^18^F]FEPPA at detecting markers of neurological diseases differed by disease. [^18^F]FEPPA was effective for AD [43], with significant differences in V_t_ values in many regions of the brain when comparing healthy control and disease groups. However, Strafella and co-workers found no clear association between factors of PD, such as disease duration and disease status, and the V_t_ value of [^18^F]FEPPA [44]. For first-episode psychosis, similar results were reported by Romina Mizrahi and co-workers [45].

#### 3.1.2. [^18^F]Fluoromethyl-PBR28

[^18^F]Fluoromethyl-PBR28 (**4**), (N-(2-[^18^F]fluoromethoxybenzyl)-N-(4-phenoxypyridine)-3-yl)acetamide) is a derivative similar to [^18^F]FEPPA in which the -(CH_2_)_2_-^18^F group is replaced by the -(CH_2_)^18^F group. In 2014, Lee and co-workers reported a two-step reaction to prepare [^18^F]fluoromethyl-PBR28 (Figure 1). In this study, the -CH_2_-^18^F group was substituted for the ^11^CH_3_ group in [^11^C]PBR28 to yield [^18^F]fluoromethyl-PBR28 [125].

Alkylation of desmethyl PBR28 (**9**) with 1-(chloromethyl)-3-methyl-4-phenyl-1H-1,2,3-triazol-3-ium triflate successfully produced triazolium triflate-PBR28 (**10**). Fluoromethyl-PBR28 (**4**) was prepared by reactions between tetra-n-butylammonium fluoride (TBAF)-3H_2_O and triazolium triflate-PBR (**10**) (Figure 1). Overall, the reaction had a decay-corrected radiochemical yield of 35.8 ± 3.2% (*n* = 11), and the radioactivity of [^18^F]fluoromethyl-PBR28 varied from 220 to 340 GBq/μmol.

The biological activity of fluoromethyl-PBR28 was also evaluated. The results showed that the binding affinity of fluoromethyl-PBR28 (measured by the half-maximal inhibitory concentration [IC_50_]) in the membranes of human leukocytes was 8.28 ± 1.79 nM, and the partition coefficient (log D) was 2.85 ± 0.05, which was the same as that of [^11^C]PBR28. In vitro sustainability in human serum was up to 99% after 4 h, indicating that [^18^F]fluoromethyl-PBR28 is sufficiently stable for use in vivo. In a PET study, the uptake of [^18^F]fluoromethyl-PBR28 reached the climax point at approximately 4.5 min post-injection. After 35 min, the contrast ratio of [^18^F]fluoromethyl-PBR28 between the ipsilateral and the contralateral areas had reached 3.4, which was faster than what has been seen for [^11^C]PBR.

Later, Lee and co-workers carried out a comparison study between [^18^F]fluoromethyl-PBR28 and its ^2^H derivates ([^18^F]fluoromethyl-PBR28-d_2_) [126]. The results showed that the properties of [^18^F]fluoromethyl-PBR28-d_2_ were similar to those of [^18^F]fluoromethyl-PBR28 in terms of in vitro binding affinity and lipophilicity. However, the uptake of [^18^F]fluoromethyl-PBR28-d_2_ was reduced in the skull and femur compared with [^18^F]fluoromethyl-PBR28 (1.5% ± 1.2% versus 4.1% ± 1.7% of injected dose (ID)/g at 2 h post-injection) and cleared more rapidly in the contralateral area, indicating an improved target-to-background ratio (approximately 3.8-fold for [^18^F]fluoromethyl-PBR28-d_2_ versus 3.0-fold for [^18^F]fluoromethyl-PBR28 at 30 min post-injection). Several other preclinical studies also concluded that the radiotracer [^18^F]fluoromethyl-PBR28 is a sensitive tool for diagnosing neurological diseases in animal models [46,127,128,129].

#### 3.1.3. [^18^F]FEMPA

In 2015, N-{2-[2-^18^F-Fluoroethoxy]-5-metoxybenzyl}-N-[2-(4 metoxyphenoxy) pyri-din-3-yl]axetamit ([^18^F]FEMPA) (**5**) was used as a PET tracer of TSPO in AD patients [47]. [^18^F]FEMPA underwent rapid metabolism at 20 min post-injection, and more than 20% of the tracer was detectable in the plasma, and this number decreased to 10% at 90 min. The binding affinity ratio of [^18^F]FEMPA to TSPO between HABs (2.26 ± 0.18 nM) and MABs (1.93 ± 0.75 nM and 189.8 ± 14.4 nM) was similar to that of [^11^C]PBR28. Furthermore, [^18^F]FEMPA binding to TSPO in HABs increased by an average 19.5% ± 3.0 % in AD patients when compared with controls, suggesting that [^18^F]FEMPA can detect microglial activation in patients with AD. Recently, Harding and co-workers reported on the results of a PET study using [^18^F]FEMPA to assess neuroinflammation in the cerebellum and brainstem in patients with Friedreich ataxia (FRDA) [48]. The SUV data at various positions of the brain, such as the dentate nuclei and midbrain regions, revealed an increased expression of TSPO in FRDA patients compared with healthy controls. This study also found that chronic neuroinflammation may be a critical initial pathological sign in FRDA, as shown by greater [^18^F]FEMPA uptake in patients with premature symptoms and disease stages.

#### 3.1.4. [^18^F]FEDAA1106

In 1999, Tomisawa and co-workers reported on N-(2,5-dimethoxybenzyl)-N-(5-fluoro-2-phenoxyphenyl)acetamide (DAA1106), a potential and selective TSPO radioligand [49]. [^11^C]DAA1106 was synthesized and employed as a PET radioligand [50]. Subsequent studies developed new ^18^F-radiolabeled TSPO ligands, including [^18^F]FEDAA1106 (**6**) and [^18^F] DAA1106 (**7**), by replacing the ^11^C on DAA1106 with ^18^F.

In 2003, Suzuki and co-workers reported that the synthesis of N-(5-fluoro-2-phenoxyphenyl)-N-(2-[^18^F]fluoroethyl-5-methoxybenzyl)acetamide ([^18^F]FEDAA1106) (**6**) could be achieved through a reaction between a desmethyl substrate N-(5-fluoro-2-phenoxyphenyl)-N-(2-hydroxy-5-methoxybenzyl) acetamide (DAA1123) (**11**) and the respective fluoroethylated intermediates, which were synthesized by treating [^18^F]F^−^ with CH_2_I_2_ or 2-bromoethyl triflate (BrCH_2_Ch_2_OTf) with 75% yield (Figure 2) [51]. After high-performance liquid chromatography (HPLC) purification, the radiochemical purity of the ^18^F-radiolabeled ligand was >98%, and the activity was >120 GBq/μmol. After maintaining it at 25 °C for 4 h, the radiochemical purities of [^18^F]FEDAA1106 were still >95% [51].

When Zhang and co-workers used [^18^F]FEDAA1106 for an in vivo study, they discovered the properties of [^18^F]FEDAA1106, including its binding affinity (K_i_ = 0.078 ± 0.01 nM) and lipophilicity (logD = 3.81). In this study, high radioactivity levels (2.2%−4.9% of ID/g) were found in the mouse brain, which were approximately 1.3–1.6 times greater than those of [^11^C]DAA1106 and 2–3 times greater than those of [^11^C](R)–PK 11,195 [52]. The radioactivity of [^18^F]FEDAA1106 was the highest in the olfactory bulb region, followed by the cerebellum, while the frontal cortex showed low uptake [51]. Subsequent attempts have been made to develop simpler and more efficient methods of synthesizing [^18^F]FEDAA1106, as well as preclinical and clinical studies.

In 2014, Cuhlmann and co-workers reported that [^18^F]FEDAA1106 could be used in an in vivo study for the detection and mapping of vascular inflammation using PET imaging [53]. This finding suggests that [^18^F]FEDAA1106 can bind at high specificity to TSPO tissues.

Several clinical studies using [^18^F]FEDAA1106 have been carried out to investigate neurological diseases. Takano and co-workers found no major differences in the distribution volumes of [^18^F]FEDAA1106 between MS patients and healthy controls [54]. No increased binding in AD occurred in an in vivo PET study of TSPO with [^18^F]FEDAA1106 [55]. However, these studies involved a small number of patients and did not consider other factors, including TSPO genotype polymorphisms. More detailed and larger scale clinical studies are therefore needed.

#### 3.1.5. [^18^F]DAA1106

Historically, it has proven difficult to produce TSPO ligands with radioisotopes directly attached to a benzene ring, and the process often results in poor synthesis efficiency and radioactivity, making TSPO ligands unsuitable for PET imaging.

In 2007, Zhang and co-workers proposed a new synthetic method using intermediate diphenyl iodonium salt to produce a new TSPO ligand, N-(2,5-dimethoxybenzyl)-N-(5-[^18^F]fluoro-2-phenoxyphenyl)acetamide ([^18^F]DAA1106) (7). However, [^18^F]DAA1106 could not be prepared through this procedure for clinical use due to the instability of the diphenyl iodonium precursor and its unsuitability for automated manufacturing [56].

Later, Zhang and co-workers reported a novel method for the synthesis of [^18^F]DAA1106 from N-(2,5-dimethoxybenzyl)-N-(5-iodo-2-phenoxyphenyl)acetamide, which was prepared according to a previous procedure [57]. 4-Bromo-1-flouro-2-nitrobenzene (**12**) was transformed to N-(2,5-dimethoxybenzyl)-N-(5-iodo-2- phenoxyphenyl)acetamide (**13**) in six steps, and then, N-(2,5-dimethoxybenzyl)-N-(5-iodo-2- phenoxyphenyl)acetamide (**13**) was converted to a precursor with spirocyclic iodonium ylide (SCIDY) (**14**) through two steps. The SCIDY was then radiofluorinated with [^18^F]F^−^ to give [^18^F]DAA1106 (**7**) (Figure 3) [58].

The molar activity of [^18^F]DAA1106 was 60–100 GBq/μmol, and the radiochemical purity was > 98%. After being kept at room temperature for 120 min, the radiochemical purity of [^18^F]DAA1106 remained above 95% [58].

When preclinical evaluation studies using [^18^F]DAA1106 were performed, [^18^F]DAA1106 showed considerable absorption in the brain (> 1.5% ID/g) but low uptake in bone. PET imaging studies in ischemic rat brains indicated that the uptake of [^18^F]DAA1106 on the ipsilateral side was higher than on the contralateral side. The ischemia regions could be seen clearly, and the uptake ratio between the ipsilateral and contralateral sides was 1.9 ± 0.3. A PET study validated the distribution of [^18^F]DAA1106 in the ischemic rat brain and its remarkable selectivity for TSPO [58].

#### 3.1.6. [^18^F]PBR06

^18^F-N-fluoroacetyl-N-(2,5-dimethoxybenzyl)-2-phenoxyaniline ([^18^F]PBR06) (**8**) is another potent PET tracer. [^18^F]PBR06 (**8**) was synthesized from 2,5-dimethoxybenzaldehyde as a starting material by Pike and co-workers in 2009 [130]. A bromo precursor (**16**) was synthesized from 2,5-dimethoxybenzaldehyde (**15**) in three synthetic steps and then radiolabeled with ^18^F to yield [^18^F]PBR06 (**8**) (Figure 4).

The prepared [^18^F]PBR06 had high radiochemical purity (100%), and the molar activity was 3.4 to 9.0 C_i_/μmol. Preclinical studies showed that the affinities (K_i_) of PBR-06 for TSPO in the brains of rats, monkeys, and humans were 0.180 ± 0.007, 0.318 ± 0.018, and 0.997 ± 0.070, respectively, similar to those of [^3^H]PK11195. The measured log D value was 4.05 ± 0.02, and [^18^F]PBR06 exhibited long-term stability in monkey whole blood and plasma, as well as in a sodium phosphate buffer (0.15 M, pH 7.4).

[^18^F]PBR06 has been used to examine neurological diseases associated with neuroinflammation [59,60]. Billy and co-workers evaluated [^18^F]PBR06 in a mouse model of stroke-induced neuroinflammation, and their PET study suggested that infarct areas had greater [^18^F]PBR06 uptake. The buildup of [^18^F]PBR06 in infarct regions was more gradual than in non-infarct regions, peaking at 10 min and then decreasing slowly to the point where it was greater than in the non-infarct regions after 15 min. By 60 min, the uptake increase in infarct regions had gradually stabilized and peaked at a level, which was 65% greater than in non-infarct regions. A displacement and pre-blocking study of unlabeled PK11195 competitively inhibited [^18^F]PBR06, demonstrating that [^18^F]PBR06 specifically binds to TSPO [59].

A PET study using [^18^F]PBR06 to measure treatment response in HD reported by Simmons and co-workers [60] demonstrated that [^18^F]PBR06 was sensitive enough to detect increased TSPO, which is a symptom of HD in mouse models. In particular, [^18^F]PBR06 could detect an increased TSPO status at two disease phases, including the early symptomatic phase, in mice with HD, as well as the ameliorative effects of LM11A-31 chemotherapy. These findings indicate that [^18^F]PBR06 can be an effective PET tracer in clinical HD investigations.

### 3.2. Pyrazolopyrimidines

Pyrazolopyrimidine compounds were developed based on the basic pyrazolo [1,5-a] pyrimidine skeleton, including a hetero bicyclic compound containing a pyrazole fused to a pyrimidine ring. This structure has been shown to interact with a wide variety of biological targets, making it a privileged scaffold in pharmaceutical chemistry [131]. Compounds with a pyrazolopyrimidine moiety have proven to have strong affinity for TSPO. N,N-Diethyl-2-[2-(4-methoxyphenyl)-5,7-dimethyl-pyrazolo [1,5-a]pyrimidin-3-yl]-acetamide (DPA-713) is a potential TSPO ligand [132]. DPA-713 has a greater affinity (K_i_ = 4.7 nM) for TSPO compared with PK11195 (K_i_ = 9.3 nM) and is significantly more selective for TSPO when compared with the central benzodiazepine receptor (CBR), which has a K_i_ > 10,000 nM. Subsequent studies of ^11^C-radiolabeling of DPA-713 produced a useful PET tracer, which has been used in several preclinical and clinical studies [132,133]. Based on positive results, new ^18^F-radiolabeled TSPO ligands, including [^18^F]DPA714 and [^18^F]DPA, were developed (Figure 3).

#### 3.2.1. [^18^F]DPA-714

[^18^F]DPA-714 (**17**) is a ^18^F-labeled compound with a pyrazolopyrimidine group, as reported in 2008 by Kassiou and co-workers. [^18^F]DPA-714 (**17**) was synthesized by a seven-step process (Figure 5) [66]. First, desmethyl DPA713 (**22**) was prepared in five steps from methyl 4-methoxylbenzoate (**21**) [133]. The treatment of desmethyl DPA713 (**22**) with toluene-4-sulfonic acid 2-hydroxy-ethyl ester and triphenylphosphine then gave a tosylate precursor (**23**). Finally, after ^18^F-radiolabeling process, [^18^F]DPA-714 (**17**) was generated with a 16% radiochemical yield and a specific activity of 270 GBq/μmol. DPA-714 displayed an affinity for TSPO of K_i_ = 7.0 nM, which was higher than that of PK11195 (K_i_ = 9.3 nM). The log D value for DPA-714 was 2.44, which was similar to that for DPA713 and less than that for PK11195 (3.35). A PET study using rats with quinolinic acid lesions reported increased uptake in lesion-containing areas of the brain, indicating that [^18^F]DPA-714 can cross the blood–brain barrier. Biodistribution and specificity studies of baboons confirmed that PK11195 also inhibited binding of [^18^F]DPA-714 to TSPO in baboon brains, indicating that [^18^F]DPA-714 can bind specifically to TSPO [66].

A study of the pharmacological properties of [^18^F]DPA-714 in the brains of monkeys was carried out by Lavisse and co-workers in 2015 [67]. The results showed that [^18^F]DPA-714 was widely distributed in the brain but concentrated mainly in the hippocampus, occipital cortex, and, to a lesser extent, the cerebellum. The degree of association of [^18^F]DPA-714 with TSPO in the brain was approximately 73%, demonstrating the high specificity of [^18^F]DPA-714 to TSPO in both normal and neurodegeneration-induced models.

Ribeiro and co-workers used [^18^F]DPA-714 in PET imaging to evaluate post-stroke neuroinflammation. They found that enhanced uptake of [^18^F]DPA-714 co-localized with infarct tissue. The injured tissue exhibited different [^18^F]DPA-714 kinetics compared with healthy tissue, suggesting that [^18^F]DPA-714 can be used to determine the degree of neuroinflammation in acute strokes [68].

Remy and co-workers reported that the V_t_ value of [^18^F]DPA-714 in the brain was higher in HABs (45.9% ± 4.8%) than in MABs, suggesting that the genotyping polymorphism of TSPO affects [^18^F]DPA-714 binding [69].

However, clinical studies in AD patients using [^18^F]DPA-714 did not yield verifiably positive results [70]. The results of an analysis of brain areas indicated that there are no statistically significant differences between the volume of distribution V_t_ and the binding potential (BP) between the subjects with and without the disease, demonstrating that [^18^F]DPA-714 may not be useful for early detection of AD.

In 2020, Backhaus and co-workers, who chose [^18^F]DPA-714 to carry out a clinical study of primary angiitis of the central nervous system (PACNS), found that reduced [^18^F]DPA-714 uptake was observed after anti-inflammatory treatment in patients with PACNS [71]. This confirmed that [^18^F]DPA-714 is suitable for the diagnosis and treatment monitoring of PACNS.

#### 3.2.2. [^18^F]VUIIS-1008

In an attempt to identify pyrazolopyrimidine derivatives that produce more effective TSPO ligands with higher affinity to TSPO, 2-(5,7-diethyl-2-(4-(2-[^18^F]fluoroethoxy)phenyl)-pyrazolo [1,5-a]pyrimidin-3-yl)-N,N-diethylacetamide (^18^F-VUISS1008) (**18**), a novel TSPO ligand, was created by Manning and co-workers in 2013 [78].

[^18^F]VUIIS1008 (**18**) was prepared using a microwave-assisted organic synthesis method. 3-(4-Methoxyphenyl)-3-oxopropanenitrile (**24**) was used as a starting material to yield pyrazole skeleton, followed by condensation with substituted diones to give the tosylate precursor (**25**). ^18^F-radiolabeling of the tosylate precursor produced [^18^F]VUIIS1008 (**18**) (Figure 6).

Molecularly, VUIIS1008 is structurally similar to DPA714, with only a slight structural change, in which the two dimethyl groups on the pyrazolo [1,5-a]pyrimidine framework are replaced by two diethyl groups. The introduction of the ethyl groups increases lipophilicity. The value for lipophilicity (at pH = 7.5) of VUISS-1008 is 2.50, indicating that VUISS-1008 can penetrate membranes and bind intracellular targets, such as TSPO. Ethyl group modification led to a surprising improvement in affinity. The K_i_ value of VUISS-1008 to TSPO is subnanomolar (0.27 nM), which is significantly superior to that of DPA714 (9.73 nM) and PBR28 (4.0 nM). VUISS-1008 also has a low affinity for CBR (K_i_ > 10,000 nM), suggesting strong selectivity for TSPO. Preclinical imaging and distribution studies have demonstrated that the uptake of [^18^F]VUIIS1008 in the brain is largely confined to the tumor, with only minor accumulation in neighboring tissues. The tumor-to-normal brain distribution volume ratio (V_t_) is reportedly 6.0, providing effective contrast between tumor and normal tissue images.

Continuing their efforts to develop novel TSPO ligands based on a DPA714 structure in 2017, Manning and co-workers reported two new TSPO ligands, [^18^F]VUIIS1009A and [^18^F]VUIIS1009B [79]. Each one is a regioisomer of the other, in which the methyl group and ethyl group positions on the pyrazolo [1,5-a]pyrimidine framework are exchanged. Both VUIIS1009A and VUIIS1009B have high TSPO binding affinity, with measured IC_50_ values 1/500th that of DPA-714 (IC_50_ = 14.4 pM for VUIIS1009A and IC_50_ = 19.4 pM for VUIIS1009B). In a PET study using rats with C6 gliomas, both [^18^F]VUIIS1009A and [^18^F]VUIIS1009B accumulated in higher concentrations in the tumor tissue compared with [^18^F]DPA-714. [^18^F]VUIIS1009B in particular exhibited considerably high tumor uptake (V_t_) compared with that of [^18^F]VUIIS1009A.

#### 3.2.3. [^18^F]DPA-C5yne

DPA-C5yne (N,N-diethyl-2-(2-(4-(3-fluoropent-1-yn-1-yl)phenyl)-5,7 dimethylpyrazolo [1,5-a]pyrimidin-3-yl)acetamide) (**19**) is another prominent derivative of DPA-714, in which a fluoroalkyn-1-yl group had been substituted for the fluoroethoxy group to connect to a phenylpyrazolopyrimidine skeleton. The preparation of [^18^F]DPA-C5yne (**19**) was achieved through the ^18^F-labeling process from tosylate (6-(4-(3-(2-(diethylamino)-2-oxoethyl)-5,7-dimethylpyrazolo [1,5-a]pyrimidin-2-yl)phenyl)pent-4-yn-1-yl) 4-methylbenzenesulfonate) (**27**), which was produced from methyl 4-iodobenzoate (**26**) in a six-step process (Figure 7) [80].

[^18^F]DPA-C5yne has a high affinity for TSPO (K_i_ = 0.35 nM) and strong lipophilicity (Log P = 2.39). Moreover, [^18^F]DPA-C5yne is stable in plasma for at least 90 min at 37 °C, making it a promising PET tracer for targeting TSPO [80].

In 2014, Damont and co-workers used [^18^F]DPA-C5yne in a PET study. [^18^F]DPA-C5yne could clearly detect a lesion in a rat’s brain, with strong differences visible between the lesioned site and the contralateral hemisphere’s equivalent site [81]. Although the uptake of [^18^F]DPA-C5yne in the lesioned area was less than that of [^18^F]DPA-714 (0.24 % ID/mL for [^18^F]DPA-C5yne and 0.30 % ID/mL for [^18^F]DPA-714), the uptake of [^18^F]DPA-C5yne in the background area was only 0.05% ID/mL, which was considerably lower compared with the 0.08% ID/mL of [^18^F]DPA-714. The contrast ratio of [^18^F]DPA-C5yne was therefore significantly more influential than that of [^18^F]DPA-714.

#### 3.2.4. [^18^F]F-DPA

N,N-Diethyl-2-(2-(4-fluorophenyl)-5.7-dimethylpyrazolo [1,5-a]pyrimidine-3-yl)acetamide ([^18^F]F-DPA) (**20**) is another promising TSPO PET ligand containing pyrazolopyrimidine moiety. Although [^18^F]F-DPA (**20**) has a molecular structure that closely resembles [^18^F]DPA-714, the fluorine atom in [^18^F]F-DPA (**20**) is bonded directly to the phenyl ring without an intermediate alkyl or alkoxy group, which improves the metabolic stability of the radiotracer.

[^18^F]F-DPA (**20**) was developed by Annelaure and co-workers [82]. Initially, [^18^F]F-DPA (**20**) was prepared using radiofluorination reactions and aryltrimethylammonium salt and diaryliodomium salt as precursors. However, this protocol provided a low radiochemical yield (<3%), even though [^18^F]F-DPA (**20**) showed strong affinity for TSPO (K_i_ = 1.7 nM) and high selectivity with the CBR (K_i_ > 1 μM).

In 2017, Solin and co-workers reported that [^18^F]Selectfluor could be employed to radiolabel F-DPA (**20**). First, methyl 4-iodobenzoate (**26**) was converted to N,N-Diethyl-2-(2-(4-(tributylstannyl)phenyl)-5,7-dimethylpyrazolo [1,5-α]pyrimidin-3-yl)acetamide (**28**) in a five-step process. Then, N,N-Diethyl-2-(2-(4-(tributylstannyl)phenyl)-5,7-dimethylpyrazolo [1,5-α]pyrimidin-3-yl)acetamide (**28**) was radiolabeled by [^18^F]Selectfluor-bis(triflate) to produce [^18^F]F-DPA (**20**) (Figure 8). In this radiofluorination, [^18^F]F-DPA was generated with a 15% ± 3% decay-corrected radiochemical yield and a low specific activity (7.8 ± 0.4 GBq/μmol). The radiochemical purity was greater than 99% [83].

An in vivo study showed that [^18^F]F-DPA was rapidly taken up in the brain and achieved equilibrium 20–30 min post-injection. At 90 min post-injection, the radioactivity levels of [^18^F]F-DPA in the plasma and brain were 28.3 ± 6.4 % and 93.5 ± 2.8 %, respectively. These findings indicate that [^18^F]F-DPA is a promising TSPO radiotracer [83].

In 2022, a study to evaluate the ability of [^18^F]F-DPA PET to detect microglial activation of neuropathic pain was carried out by Lida and co-workers. In this study, [^18^F]F-DPA was superior to [^11^C]PK11195 in aiding the imaging of spinal cord inflammatory locations in model rats on ex vivo autoradiography. The [^18^F]F-DPA uptake of the vertebral body was twice that of bone from the skull. However, PET scanners were not able to detect enhanced absorption of [^18^F]F-DPA at the inflammation site in this model [84].

### 3.3. Imidazopyridine Acetamides

Alpidem is an anxiolytic drug, which has been identified as a potential and selective TSPO ligand [134]. [^11^C]CLINME was prepared from alpidem and employed as a PET tracer [134,135,136,137,138]. These two classes of compounds are structurally similar and are classified as imidazopyridine acetamides. This group of substances has the basic skeleton of imidazo [1,2-a]pyridines, which contain nitrogen heterocycles. Nitrogen heterocycles have been shown to meet most of the major criteria when developing new drugs with pharmacological promise, including biological activity, solubility, and other properties [139]. Subsequent studies developed new ^18^F-radiolabeled imidazopyridine acetamides by replacing ^11^C on alpidem and CLINME with ^18^F (Figure 4).

#### 3.3.1. PBR102 and PBR111

In 2008, Greguric and co-workers developed new and selective ^18^F-radiolabeled ligands based on imidazo [1,2-a]pyridineacetamides [87]. In this study, fluoroethoxy and fluoropropoxy substituents were attached to the 4′-position of a 2-phenyl ring, generating PBR compounds (fluoroethoxy for [^18^F]PBR102 (**29**) and fluoropropoxy for [^18^F]PBR111 (**30**)) with superior characteristics for PET.

A p-toluenesulfonyl precursor (**35**) of [^18^F]PBR102 and [^18^F]PBR111 was synthesized in a nine-step process using 5-chloropyridin-2-amine (**34**) as a starting material. Radiofluorination was conducted using a substitution reaction of the p-toluenesulfonyl group by the [^18^F]fluoride group in the precursors in the presence of K_2.2.2_ and K_2_CO_3_ to give [^18^F]PBR102 (**29**) and [^18^F]PBR111 (**30**) (Figure 9). The products were obtained with a 55–75% radiochemical yield after HPLC purification and were >95% pure.

Subsequent studies of biological properties showed that imidazopyridineacetamide derivative ligands had high selectivity and binding affinity for TSPO. The binding affinities of PBR102 and PBR111 were 5.8 ± 0.4 nM and 3.7 ± 0.4 nM, respectively. Their lipophilicities were appropriate for brain uptake (log P = 2.7 ± 0.1 for PBR102 and 3.2 ± 0.1 for PBR111). The highest brain uptake was 0.2% ID/g for [^18^F]PBR102 and 0.4% ID/g for [^18^F]PBR111 for the first hour. Biodistribution studies showed that radioactive concentrations in the brain had kinetics similar to that of blood, indicating a balance between brain and blood. Preclinical studies showed that [^18^F]PBR102 had a binding affinity of K_i_ = 15.5 ± 5.3 nM for HABs and a binding affinity of K_i_ = 56.3 ± 6.5 nM for LABs (in human platelets), such that a LAB/HAB ratio of 3.6 was similar to the ratio for [^18^F]PBR111 (R = 4.0). However, the K_i_ values for LABs were large enough to quantify TSPO binding with both radiotracers, regardless of the TSPO polymorphism [88].

Later, Hobson and co-workers used [^18^F]PBR111 to evaluate neuroinflammation status by diisopropylfluorophosphate poisoning in rats [89]. A significant difference in PET images using [^18^F]PBR111 was observed before and after 3, 7, 14, 21, and 28 days of toxicity. At days after exposure, the SUV data exhibited a marked change. In comparison with the control group, [^18^F]PBR111 uptake in the piriform cortex, hippocampus, thalamus, and amygdala was enhanced. The neuroinflammation observed by PET using [^18^F]PBR111 was significantly associated with seizure severity over the first 4 h post-diisopropylfluorophosphate intoxication. [^18^F]PBR111 was shown to be a reliable, non-invasive technique for monitoring neuroinflammation by organophosphates intoxication.

#### 3.3.2. PBR316

Recently, it has been reported that [^18^F]PBR316 (**31**) can advantageously image target proteins [93]. Structurally, the [^18^F]PBR316 (**31**) skeleton is similar to that of [^18^F]PBR102 and [^18^F]PBR111. However, the methylene group of the 3-acetamide chain is replaced by a carbonyl group, and the N,N′-diethyl group is changed to an N,N′-dimethyl group.

The precursor (**37**) of [^18^F]PBR316, a tosylate derivative, was prepared using 4-acetylphenethyl acetate (**36**) as a starting material in a nine-step process (Figure 10). The tosylate precursor (**37**) was then radioflourinated to produce [^18^F]PBR316 (**31**) in a radiochemical yield of 20 ± 5% (*n* = 9), with high radiochemical purity (>99%) and 400 GBq/μmol of molar activity.

The results of an in vitro study indicate that PBR316 had high affinity and selectivity with TSPO (K_i_ = 6.0 ± 1.4 nM). PBR316 was proven to outperform other imidazopyridine ligands, including PBR111 and PBR102, in terms of binding selectivity between TSPO and CBR. Furthermore, the lipophilicity value (log P) was 2.16 ± 0.07, which was suitable for brain uptake. A PET study indicated that [^18^F]PBR316 showed good uptake in the heart, endocrine tissue, and kidneys (in rats), and [^18^F]PBR316 could cross the brain–blood barrier and be absorbed by the brain and in the area of the olfactory bulb (6.9 ± 0.4% ID/g at 4 h). This ratio was larger than those of both [^18^F]FEDAA1106 and [^18^F]PBR111.

#### 3.3.3. [^18^F]CB251

In 2016, Perrone and co-workers developed a novel ^18^F-labeled TPSO ligand ([^18^F]CB251) (**32**) from a 6,8-di-substituted imidazo [1,2-a]pyridine-N,N-dipropylacetamide structure [94].

The starting material was 4-(4-methulxyphenyl)-4-oxobutanoic acid (**38**). Using seven reaction steps, it was converted to a tosylate precursor (**39**) of [^18^F]CBR251 [95]. Finally, radiosynthesis was conducted to give the desired product, [^18^F]CB251 (**32**) (Figure 11). After HPLC purification, [^18^F]CB251 with a radiochemical yield of 11.1% ± 3.5% (*n* = 14) was obtained, and the radioactivity was 104 to 154 GBq/μmol.

The evaluation of the biological activity of [^18^F]CBR251 indicates that [^18^F]CBR251 has high binding affinity and selectivity for TSPO (K_i_ = 0.27 ± 0.09 nM), which is better than those of PK 11,195 and PBR28. Furthermore, the partition coefficient (log D) of [^18^F]CBR251 was 3.00 ± 0.03, which suggests that it can easily cross the blood–brain barrier. At 5 min post-injection, the brain uptake of [^18^F]CB251 was 2.89 ± 0.23 %ID/g.

Recently, Youn and co-workers proved the usefulness of [^18^F]CBR251 for visualizing neuroinflammation, regardless of the polymorphism of TSPO [96]. The study confirmed that [^18^F]CB251 is selective for TSPO and absorbed by immune cells, which are activated by high TSPO expression. These results show that [^18^F]CB251 can be used to image TSPO-related neuroinflammation. A PET study demonstrated that [^18^F]CB251 can be used to examine the treatment effect of anti-inflammatory medicine in a lipopolysaccharide-induced neuroinflammation mouse model.

#### 3.3.4. [^18^F]BS224

The rs6971 polymorphism on TSPO causes an amino acid substitution (Ala147Thr) in the protein’s fifth transmembrane loop. TSPO polymorphisms reportedly affect binding to TSPO ligands, limiting their application as biomarkers in PET. In 2021, Kim and co-workers successfully developed a new TSPO PET ligand, 2-(-2-(4-[^18^F] fluorophenyl)-6,8-dichloro-imidazo [1,2-a] pyridin-3-yl)-N, N-dipropylacetamide ([^18^F]BS224) (**33**), which was insensitive to rs6971 polymorphisms [97].

[^18^F]BS224 was synthesized using aromatic ^18^F-fluorination. The precursors were synthesized from methyl-4-iodobenzoate (**40**) in a four-step pathway using a boronic acid pinacol ester derivative (**41**) or a six-step process with iodotoluene tosylate derivatives (**42**). The precursors were then treated with 18-Crown-6 and Cs^18^F to produce [^18^F]BS224 (**33**) (Figure 12). After the purification process, the [^18^F]BS224 ligand was obtained at a decay-corrected radiochemical yield of 39% ± 6.8% (*n* = 8) and a radiochemical purity of >99%.

The biological characteristics of CB251 may still exist in BS224. For example, in vivo experiments demonstrated that BS224 has a strong affinity and specificity for TSPO compared with CBR (K_i_ = 0.51 ± 0.03 nM for TSPO and K_i_ > 105 for CBR), which were similar to those of the CB251 ligand. The partition coefficient value (log D) of [^18^F]BS224 was 2.78 ± 0.04, and a stability test of [^18^F]BS224 showed that >99% of the initial compound remained intact for 120 min in serum. [^18^F]BS224 was also stable in the brain, heart, kidneys, and lungs. The ratio of the binding affinity of the ligand (LAB/HAB) was 0.76, demonstrating that BS224 is not sensitive to the polymorphism of TSPO. Furthermore, in PET studies using lipopolysaccharide-induced inflammation and ischemic-stroke rat models, [^18^F]BS224 produced clear images of inflammatory lesions with a good contrast ratio between damaged tissue and background without skull uptake.

### 3.4. Oxopurine

AC-5216, an antianxiety and antidepressant-like drug, was found to be a potential selective TSPO ligand in 2004 [140]. [^11^C]AC-5216 was synthesized and demonstrated to be a useful PET radioligand [141]. Subsequent studies developed a new ^18^F-radiolabeled TSPO ligand ([^18^F]FEDAC) based on the structure of AC-5216 by modifying the substitute groups and replacing ^11^C with ^18^F (Figure 5).

In 2009, N-benzyl-N-methyl-2-[7,8-dihydro-7-(2-^18^F-fluoroethyl)-8-oxo-2-phenyl-9H-purin-9-yl]acetamide (^18^F-FEDAC) (**43**) was developed as a new TSPO ligand by Zhang and co-workers [98]. [^18^F]FEDAC (**43**) was synthesized in a four-step process (Figure 13). The precursors (**45**) for radiosynthesis were synthesized from a glycine derivative (**44**) in three steps. A radiolabeling procedure using reactions of the radioactive intermediate [^18^F]FCH_2_CH_2_Br and a precursor (**45**) gave [^18^F]FEDAC (**43**) with a yield of 69%. The radiochemical purity was >98%, and the specific activity was 30–95 GBq/μmol.

Several preclinical and clinical studies have been performed to evaluate the properties and potential applications of [^18^F]FEDAC. The studies reported that [^18^F]FEDAC exhibits suitable properties for a PET tracer, such as an inhibition constant (K_i_) of 1.3 nM and a powerful binding affinity for TSPO with high selectivity (K_i_ = 8.700 nM for CBR). In a PET study using a neuroinflammatory rat model, [^18^F]FEDAC absorbed substantial amounts of radioactivity in the kainic acid-infused striatum, an area of the brain associated with high levels of TSPO expression.

Later, [^18^F]FEDAC was used as PET imaging agent for activated macrophages. In a study conducted by Cheon and co-workers in 2018, PET imaging with ^18^F-FEDAC was employed to predict the therapeutic benefits of biological disease-modifying anti-rheumatic medications with anti-inflammatory properties to suppress active macrophages [99]. Similarly, Yamashita and co-workers demonstrated that [^18^F]FEDAC could effectively visualize atherosclerotic lesions in rabbits and humans. The results showed that, in wounded arteries, the SUV of [^18^F]FEDAC was 0.574 ± 0.24, which was greater than that reported in uninjured arteries (0.277 ± 0.13) or the myocardium (0.189 ± 0.07) [100].

### 3.5. Acetamidobenzoxazolone

[^11^C]MBMP was synthesized as a potential and selective TSPO ligand and employed in a PET study [102]. Subsequent studies developed new ^18^F-radiolabeled TSPO ligands, such as [^18^F]FEBMP and [^18^F]FPBMP (Figure 6).

#### [^18^F]FEBMP and [^18^F]FPBMP

In 2014, 2-[5-(4-[^18^F]fluoroethoxyphenyl)- ([^18^F]FEBMP) and 2-[5-(4-[^18^F]fluoropropyloxyphenyl)- ([^18^F]FPBMP) -2-oxo-1,3-benzoxazol-3(2H)-yl]-N-methyl-N-phenylacetamide were reported by Zhang and co-workers [103]. [^18^F] FEBMP (**46**) and [^18^F]FPBMP (**47**) were both synthesized from 2-nitro-4-bromophenol (**48**). During the six-step procedure, 2-nitro-4-bromophenol (**48**) was converted to 2-[5-(4-Hydroxyphenyl)-2-oxo-1,3-benzoxazol-3(2H)-yl]-N-methyl-N-phenylacetamide (**49**), which reacted with 1-bromo-2-[^18^F]fluoroethane or 1-bromo-3-[^18^F]fluoropropane to yield [^18^F]FEBMP (**46**) and [^18^F]FPBMP (**47**) (Figure 14). The radiochemical yields of [^18^F]FEBMP and [^18^F]FPBMP were 22 ± 4% (*n* = 8) and 5 ± 2% (*n* = 5), respectively. The radiochemical purities for [^18^F]FEBMP and [^18^F]FPBMP were both found to be 98%, and the specific activity ranged from 98 to 364 GBq/mol.

Three-dimensional pharmacophore assessment and docking studies indicated that both molecules had a high affinity for TSPO. In vitro binding experiments with TSPO revealed that the binding affinities for FEBMP and FPBMP were 6.6 ± 0.7 nM and 16.7 ± 2.5 nM, respectively.

An ischemic rat brain image obtained from in vitro autoradiography exhibited much more binding on the ipsilateral side compared with the contralateral side. In a dynamic PET imaging study, the biodistribution of both compounds in mice suggested that regional radioactivity in the brain peaked at 0–4 min for both ligands, comparable to (R)-[^11^C]PK11195. These results suggest that both [^18^F]FEBMP and [^18^F]FPBMP are potentially useful PET ligands.

### 3.6. Pyridazinoindoles

#### [^18^F]SSR180575

SSR180575 was identified as a treatment to promote neuronal survival [142], an anti-apoptotic agent [143], and later, a potential and selective TSPO ligand [144,145,146]. [^11^C]SSR180575 was synthesized and employed as a PET radioligand [147,148,149]. Subsequent studies developed new ^18^F-radiolabeled TSPO ligands, such as [^18^F]SSR180575] (Figure 7).

Manning and co-workers developed 7-chloro-N,N-5-trimethyl-4-oxo-3(6-[^18^F]fluoropyridin-2-yl)-3,5-dihydro-4H-pyridazino [4,5-b]indole-1-acetamide, or [^18^F]SSR180575 (**50**), as a potential TSPO ligand [106].

Using ethyl 6-chloroindoline-2-carboxylate (**53**) as a starting material, [^18^F]SSR180575 precursors (**54**) were prepared in four steps to introduce potential substituents onto a benzene ring, which would enable radiolabeling with ^18^F. The radiolabeling process was then conducted to produce the final product (**50**) (Figure 15). The radiochemical purity was greater than 99%, and the decay-corrected radiochemical yield ranged from 9.3 to 19.3% (*n* = 9), with specific activities up to 5559 C_i_/mmol (206 TBq/mmol).

[^18^F]SSR180575 was evaluated in a PET study using glioma-bearing male Wistar rats. The results showed that the majority of [^18^F]-SSR180575 accumulates in tumor sites in the brain, with little accumulation in non-tumor areas. The uptake ratio between the tumor areas and non-tumor areas was greater than 10:1.

The specificity of [^18^F]SSR180575 for TSPO was investigated, and the addition of non-radioactive SSR180575 decreased [^18^F]SSR180575 uptake in tumors by 40% by comparison. This result indicates that [^18^F]SSR180575 has a high selective binding and reversibility to TSPO at the injury site.

In 2015, Damont and co-workers prepared and characterized various fluorinated derivatives of SSR180575 (**51** and **52**) (Figure 7) [107]. All of them had pyridazino [4,5-b]indole-1-acetamides moiety with minor structural modifications. Two series of compounds have been synthesized. In series 1, the substituent of the phenyl group (p-position) was modified to form eight alkyl ethers, including one methoxy derivative and seven fluorinated compounds. In series 2, the N-indole position of SSR180575 was examined with different fluoroalkyl groups replacing the methyl group, resulting in the creation of seven novel compounds.

In vitro evaluations of the binding affinities of all the prepared compounds with TSPO were carried out and compared with the SSR180575 parent compound. Eleven of the fifteen compounds exhibited similar or higher binding affinities for TSPO compared with SSR180575. The K_i_ values were in the nanomolar to subnanomolar range (0.30–8.1 nM). The selectivity of compounds to CBR was also examined. The K_i_(CBR)/K_i_(TSPO) ratio of all compounds was greater than 1000, indicating no suppression of [^3^H]flunitrazepam binding to CBR at 1 μM and suggesting that these compounds have high selectivity over CBR. The lipophilic property (log D) values ranged from 3.01 to 3.75. Although slightly higher than that of SSR180575, they were still suitable for biodistribution and brain uptake.

### 3.7. Tricyclic Indoles

#### 3.7.1. [^18^F]GE180

A tetracyclic indole structure was employed to identify novel TSPO ligands with improved properties. In 2012, Trigg and co-workers succeeded in introducing a fluoroethyl group onto indole nitrogen to create a novel TSPO ligand, [^18^F]GE180 (**55**), which retained affinity for TSPO and increased resistance to oxidation (Figure 8) [108].

For the synthesis of [^18^F]GE180 (**55**), a mesylate precursor (**58**) was prepared in a five-step process from ethyl 3-nromo-2-hydroxycyclohex-1-ene-1-carboxylate (**57**), and radiolabeling was carried out in the presence of K_2.2.2_ in MeCN at 100 °C for 10 min to produce [^18^F]GE180 (**55**) (Figure 16). The non-decay-corrected radiosynthesis yields were 25–35%.

In a rat model, [^18^F]GE180 showed high affinity for TSPO (K_i_ = 2.4 nmol/L), and metabolic studies revealed that 94% of [^18^F]GE180 in the brain was intact at 60 min post-injection.

A study comparing the effectiveness of [^18^F]GE180 and [^11^C]-(R)-PK11195 in detecting microglial activation in an acute model of neuroinflammation was carried out by Dickens and co-workers, who reported that [^18^F]GE180 could detect activated microglia in the brain [109].

In 2015, Trigg and co-workers used [^18^F]GE180 in a preclinical model of strokes. The result indicated that uptake of [^18^F]GE180 was 24% greater in ischemic lesions and inferior by 18% in the contralateral healthy site compared with [^11^C]-(R)-PK11195. The signal-to-noise ratio was 1.5-fold greater than that of [^11^C]-(R)-PK11195. A blocking study demonstrated that [^18^F]GE180 has a high binding affinity to TSPO; after 20 min [^18^F]GE180 post-injection, unlabeled GE-180 and (R)-PK11195 induced decreases in [^18^F]GE180 uptake of 69 ± 5% and 63 ± 4%, respectively. A metabolite investigation revealed that the blood pharmacokinetics of [^18^F]GE180 were comparable to those of other TSPO tracers, such as [^11^C]DPA-713 and [^18^F]DPA-714 [110].

[^18^F]GE180 was subsequently used in a PET study of an AD mouse model. López-Picón and co-workers reported increased uptake and specific binding of [^18^F]GE180 in the whole brain and hippocampus, demonstrating the potential of [^18^F]GE180 in PET to track neuroinflammation in the course of treating AD and possibly other neurodegenerative illnesses [111].

#### 3.7.2. [^18^F]GE387

In 2019, Sephton and co-workers developed another tetracyclic indole compound, [^18^F]GE387 (**56**), as a potent PET radiotracer with minimal binding sensitivity to TSPO polymorphisms [115]. The structure of [^18^F]GE387 was nearly identical to that of GE180, but an N-ethyl group was replaced by a benzyl group.

In this study, a tosylate precursor (**60**) was synthesized in a six-step process from 2-chloro-5-methoxyaniline (**59**). The ^18^F-radiolabeling process was carried out using [^18^F]fluoride in the presence of K_2.2.2_ to give [^18^F]GE387 (**56**) (Figure 17). Supercritical fluid chromatography was used to achieve chiral separation between the two enantiomers. After the ^18^F-labeling process, the R and S enantiomers of [^18^F]GE387 were obtained with decay-corrected radiochemical yields of 21.3% ± 16.7% (*n* = 9) and 25.6% ± 7.1% (*n* = 9), respectively, and molar activities of 55.8 ± 35.6 and 63.5 ± 39.5 GBq/μmol. A PET study using [^18^F]GE387 in male Wistar rats demonstrated that racemic [^18^F]GE387 can enter the brain.

In 2021, Ramakrishnan and co-workers reported a preclinical evaluation of two enantiomers of [^18^F]GE387 [116]. PET studies using lipopolysaccharide neuroinflammation rat and monkey models indicated that the (S)-[^18^F]GE387 isomer was more efficient than the (R)-[^18^F]GE387 isomer, with rapid brain uptake. A competitive binding experiment utilizing unlabeled (S)-GE387 against [^3^H]PK11195 produced K_i_ values of 5.48 ± 0.68 nM for HABs and 9.83 ± 1.28 nM for LABs, for a LAB/HAB ratio of 1.8. PET using (S)-[^18^F]GE387 can therefore distinguish between inflamed and normal brain tissue, outperforming other popular TSPO ligands.

### 3.8. Quinoline/Isoquinoline/Quinazoline Carboxamides

[^11^C]PK11195, a first-generation TSPO ligand, has been employed for PET studies. Subsequent studies developed new ^18^F-radiolabeled TSPO ligands (Figure 9).

#### 3.8.1. Quinoline Carboxamide

In 2010 and 2013, Andrew and co-workers, who were working on developing SPECT imaging agents, created quinoline-2-carboxamide derivatives containing iodine atoms that are structurally similar to PK11195. They found that the derivatives had strong affinity to TSPO but high lipophilic properties [117,118]. In 2015, Andrew and co-workers developed a novel ^18^F-radiolabeled TSPO ligand, 3-fluoromethylquinoline-2-carboxamide ([^18^F]AB5186) (**61**), for PET studies. The addition of a less lipophilic fluorine atom was expected to be effective in PET [119].

[^18^F]AB5186 (**61**) was synthesized in a six-step process from (2-aminophenyl)(phenyl)methanone (**64**). The final step in the production of [^18^F]AB5186 was radiofluorination of a precursor (**65**) by replacing the chloride atom with ^18^F. The ^18^F-radiolabeling process was achieved by using [^18^F]-potassium fluoride in the presence of Kryptofix-222 (Figure 18). [^18^F]AB5186 (**61**) was obtained with a 38% ± 19% (*n* = 7) decay-corrected radiochemical yield. The radiochemical purity was greater than 99%, and the specific activity was 0.6 ± 0.2 C_i_/μmol.

Competition-binding experiments demonstrated that the binding affinity of AB5186 to TSPO was in the nanomolar range (K_i_ = 2.8 nM), which was similar to that of PK11195. The major physicochemical properties of AB5186 suggested that it may be a useful PET imaging tracer. AB5186 had a permeability (P_m_) of 0.5 and a plasma protein binding capacity of 89.7%, suggesting that AB5186 can cross the blood–brain barrier.

A PET study on mouse bearing an intracranial U87MG-Luc2 tumor demonstrated that [^18^F]AB5186 was clearly binding to tumors, and its location was consistent with that reported by histology. These findings demonstrate the capacity of [^18^F]AB5186 to detect TSPO in vivo under pathological circumstances.

#### 3.8.2. Isoquinoline Carboxamide

In humans, an rs6971 polymorphism results in three TSPO phenotypic binding profiles. LABs frequently respond poorly to second-generation tracers, resulting in their removal from clinical investigations. In 2019, Neydher and co-workers reported (R)-[^18^F]NEBIQUINIDE (**62**), a novel pyridinyl isoquinoline derivative for TSPO PET imaging, to improve this situation [121].

A chloride precursor (**67**) of (R)-[^18^F]NEBIFQUINIDE was synthesized from methyl 1-bromoisoquiniline-3-carboxylate (**66**) in four steps. The precursor (**65**) was then radiolabeled by [^18^F]KF/Kryptofix-222 to produce the target product (**62**) (Figure 19).

The efficacy of this PET tracer was examined in a preclinical evaluation study. The measured inhibition constant (K_i_) showed that (R)-[^18^F]NEBIQUINIDE had a higher affinity for TSPO in a low nanomolar range. Moreover, the binding ratio of [^18^F]NEBIQUINIDE with HABs and LABs was 0.93, indicating that (R)-[^18^F]NEBIQUINIDE was less affected by the polymorphism of TSPO.

When the log P value of (R)-[^18^F]NEBIQUINIDE was measured and compared to that of other tracers, it was found to be in the same range. The metabolic stability of (R)-[^18^F]NEBIQUINIDE was also screened, and, at 60 min post-injection, a very small portion of (R)-[^18^F]NEBIQUINIDE was metabolized, while 98.5 ± 0.6% of the initial tracer was still present. This indicates that (R)-[^18^F]NEBIQUINIDE was more stable than [^18^F]PBR102 and [^18^F]PBR111.

PET imaging showed that the heart and lungs, which are TSPO-rich organs, exhibit the greatest uptake of (R)-[^18^F]NEBIQUINIDE (12 ± 4% and 11 ± 4 % ID/cc, respectively). The brain showed modest absorption (1.8 ± 0.3 ID/cc). Given these results, (R)-[^18^F]NEBIQUINIDE was proposed as a potential TSPO ligand to detect neuroinflammatory diseases and tumors.

#### 3.8.3. Quinazoline Carboxamide

[^11^C]ER176 (^11^C-(R)-N-sec-butyl-4-(2-chlorophenyl)-N-methylquinazoline-2-carboxamide), a novel derivative of [^11^C]-(R)-PK11195, was discovered in 2017, demonstrating potential TSPO ligand properties for PET studies [122]. Although [^11^C]ER176 is recognized as an effective TSPO ligand in humans, subsequent studies have developed new ^18^F-radiolabeled TSPO ligands by replacing ^11^C on ER176 with ^18^F.

In 2021, Victor and co-workers developed a procedure to ^18^F-radiolabel ER176 (**63**) [123]. First, (R)-N-(sec-butyl)-N-methyl-4-oxo-3,4-dihydroquinazoline-2-carboxamide (**68**) was transformed to an aryl(mesityl)iodonium salt precursor (**69**) in a four-step process. In the radiolabeling of [^18^F]ER176, the precursor (**69**) in dimethylformamide was treated with a solution of tetrakis(acetonitrile)-copper(I) triflate, and the resulting solution was heated by microwave irradiation with a combination of [^18^F]fluoride ions and K_2.2.2_/K_2_CO_3_ solution to generate [^18^F]ER176 (**63**) (Figure 20). [^18^F]ER176 had a 21% (*n* = 3) radiochemical yield and a radiochemical purity of >95%.

A PET study using mice was carried out to evaluate [^18^F]ER176 and compare it to [^11^C]ER176. In the brain, radioactivity absorption by [^18^F]ER176 peaked at 17 min, with approximately 0.8–0.9 SUV, followed by a slow, steady drop in radioactivity.

A blocking study using pretreatment of PK11195 indicated that the inhibitory effect of [^18^F]ER176 was 82%, which is similar to that of [^11^C]ER176. These results suggest that [^18^F]ER176 with a longer radioisotopic half-life could be a potential TSPO ligand.

## 4. Conclusions

Positron emission tomography is a useful imaging method for diagnosis and therapy. Because PET provides non-invasive, real-time imaging and multi-level observation of cells, tissues, and organs, it has received great attention from both clinicians and researchers.

One of the TSPO-PET imaging method’s drawbacks is that there are still few clinical studies and in-depth investigations into the cell types accountable for the TSPO response in several brain diseases. The research works regarding TSPO in humans are often extrapolated from rodent studies. However, many recent research works suggested that the expression of TSPO in animal models (preclinical investigations) may not be the same as the outcomes obtained in humans [34,150]. The sensitivity of most second-generation radioligands to the TSPO polymorphism (rs6971), which changes the TSPO affinity for the tracers, is another potential limitation of TSPO-PET neuroimaging investigations. Moreover, several recent studies have demonstrated that the expression of TSPO is caused not only by microglia or the inflammatory response, but it is also involved in many other biological activities in the cell, such as cellular metabolism, energy homeostasis, or oxidative stress during inflammation [151].

Scientists have made efforts to develop new high affinity and selectivity TSPO ligands for PET studies of neuroinflammation and neurodegenerative illnesses because overexpression of TSPO at the site of neuropathic processes is a clear and valuable marker for diagnosis of the disease state.

Several TSPO ligands radiolabeled with ^11^C and ^18^F have met the requirements of PET studies. However, the short half-life of ^11^C (20.4 min) causes major limitations in both preclinical and clinical applications. Compared with ^11^C-labeled ligands, TSPO ligands radiolabeled with ^18^F, which has a longer half-life (109.8 min), are favored as a PET tracer. Dozens of ^18^F-labeled TSPO ligands have been generated and achieved superior performance as PET tracers for imaging of diseases related to TSPO expression.

TSPO ligands in PET studies must meet a series of requirements, including high binding affinity and high selectivity. Candidate TSPO ligands should be of moderate molecular weight (<500 Da) and suitable lipophilicity, which helps the ligands cross the blood–brain barrier and enhances brain uptake. Ease of handling in the radiolabeling process should also be considered.

Despite meeting these requirements, some candidate TSPO ligands face other challenges, including the effects of TSPO polymorphisms. Next-generation TSPO-PET tracers, such as [^18^F]FEBMP, [^18^F]PBR316, (R,S)-[^18^F]GE387, and (R)-[^18^F]NEBIFQUINIDE, which have been described as being insensitive to the single genetic polymorphism rs6971, have been discovered. However, each one faces challenges with respect to pharmacokinetics, bioavailability, and safety. More preclinical and clinical studies on various models are needed to confirm their effectiveness.

More specific and highly selective TSPO ligands that can enhance clinical study results should be developed for widespread application. The synthetic steps should be simple and more easily applied, and the ^18^F-radiolabeling method should also be more effective to give ^18^F-radiolabeled TSPO ligands high radiochemical yields and high purity.

In conclusion, ^18^F-radiolabeled TSPO ligands have shown great potential for PET imaging. Future investigations should focus on generating novel structures, simplified synthesis, and highly efficient radiolabeling. Preclinical and clinical studies should be undertaken to determine the effectiveness of these TSPO ligands.

## Data Availability

Not applicable.

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
