# Peer review of "18F-Radiolabeled Translocator Protein (TSPO) PET Tracers: Recent Development of TSPO Radioligands and Their Application to PET Study"

_pharmaceutics, 2022, doi:10.3390/pharmaceutics14112545_

Round 1

Reviewer 1 Report

This is an excellent review of the literature on TSPO radioligands. I have no suggestions.

Author Response

We would like to thank positive comments about our manuscript.

Reviewer 2 Report

Please see attached

Author Response

(Q1) For the comment, “(1) Please describe briefly the status of the TSPO targeting radiopharmaceuticals in the translational viewpoint, clinical trials and future possibility.”,

(A1) We added a table summarizing status of the TSPO targeting radiopharmaceuticals including clinical trials and future possibility. (Table 1. 18F-Radiolabeled TSPO ligands, Page 04).

(Q2) For the comment, “(2) Please describe briefly the binding enhancement functionality in each tracer. How the structural alteration juggled the binding affinity? And Why?”

(A2) We added the following sentences for structural alteration of TSPO PET tracers for binding affinity.”

The sentences: “Basically, the ring opening of the diazepine ring of Ro5-4864 will form high-affinity PBR ligands, including DAA1106. Then with a pyridine ring replacing one of the benzene rings of DAA1106, the derivatives of phenoxyarylacetamide PBR were generated. This method produced a series of compounds with reduced lipophilicity but still retaining properties such as high affinity for PBR and blood–brain barrier penetration ability [41].” were added into page 06.

The sentences: “Pyrazolopyrimidine compounds were developed based on the basic pyrazolo [1,5-a] pyrimidine skeleton, including a hetero bicyclic compounds containing a pyrazole fused to a pyrimidine ring. This structure has been shown to interact with a wide variety of bio-logical targets, making it a privileged scaffold in pharmaceutical chemistry [71].” were added into page 11.

The sentences: “These two classes of compounds are structurally similar and are classified as imidaz-opyridine acetamides. This group of substances has the basic skeleton of imid-azo[1,2-a]pyridines, which contain nitrogen heterocycles. Nitrogen heterocycles have been shown to meet most of the major criteria when developing new drugs with pharmacolog-ical promise, including biological activity, solubility, and other properties [95].” were  added into page 15.

Several references was inserted and cited:

  1. Chauveau, F.; Boutin, H.; Van Camp, N.; Dollé, F.; Tavitian, B. Nuclear imaging of neuroinflammation: a comprehensive review of [11C] PK11195 challengers. Eur. J. Nucl. Med. Mol. Imaging 2008, 35, 2304-2319.
  2. Škopić, M.K.; Bugain, O.; Jung, K.; Onstein, S.; Brandherm, S.; Kalliokoski, T.; Brunschweiger, A. Design and synthesis of DNA-encoded libraries based on a benzodiazepine and a pyrazolopyrimidine scaffold. MedChemComm 2016, 7, 1957-1965.
  3. Vanda, D.; Zajdel, P.; Soural, M. Imidazopyridine-based selective and multifunctional ligands of biological targets associated with psychiatric and neurodegenerative diseases. Eur. J. Med. Chem. 2019, 181, 111569.

(Q3) For the comment, “(3) Please include some sentences for other TSPO PET tracers excluding small molecules, like macromolecules.”,

(A3) We checked literatures to find macromolecules as TSPO PET tracers, however there was no reports about macromolecules as TSPO PET tracers.

(Q4) For the comment, “(4) In lines 47 please name the diseases with recent references.

Pharmacol Ther. 2019 Feb; 194: 44–58. doi: 10.1016/j.pharmthera.2018.09.003

BMB Rep. 2020 Jan; 53(1): 20–27. doi: 10.5483/BMBRep.2020.53.1.273

Journal of Nuclear Medicine February 2013, 54 (2) 291-298; https://doi.org/10.2967/jnumed.112.108894.”

(A4) We insert the diseases name, and several references was inserted and cited:

  1. Lee, Y.; Park, Y.; Nam, H.; Lee, J.W.; Yu, S.W. Translocator protein (TSPO): the new story of the old protein in neuroinflammation. BMB Rep. 2020, 53, 20.
  2. Guilarte, T.R. TSPO in diverse CNS pathologies and psychiatric disease: A critical review and a way forward. Pharmacol. Ther. 2019, 194, 44-58.
  3. Mattner, F.; Staykova, M.; Berghofer, P.; Wong, H.J.; Fordham, S.; Callaghan, P.; Jackson, T.; Pham, T.; Gregoire, M.C.; Zahra, D.; et al. Central nervous system expression and PET imaging of the translocator protein in relapsing–remitting experimental autoimmune encephalomyelitis. J. Nucl. Med. 2013, 54, 291-298.

(Q5) For the comment, “(5) In line 56, please include the references;

Journal of Clinical Oncology 24, no. 20 (July 10, 2006) 3282-3292. DOI: 10.1200/JCO.2006.06.6068

Int. J. Mol. Sci. 2021, 22(23), 13085; https://doi.org/10.3390/ijms222313085

Annu Rev Biomed Eng. 2015 ; 17: 385–414. doi:10.1146/annurev-bioeng-071114-040723.

Int. J. Mol. Sci. 2022, 23(13), 7160; https://doi.org/10.3390/ijms23137160”

(A5) Several references was inserted and cited.

  1. Weber, W.A. Positron emission tomography as an imaging biomarker. J. Clin. Oncol. 2006, 24, 3282-3292
  2. Guan, B.; Zhou, N.; Wu, C.Y.; Li, S.; Chen, Y.A.; Debnath, S.; Hofstad, M.; Ma, S.; Raj, G.V.; He, D.; et al. Validation of SV2A-targeted PET imaging for noninvasive assessment of neuroendocrine differentiation in prostate cancer. Int. J. Mol. Sci. 2021, 22, 13085
  3. Vaquero, J.J.; Kinahan, P. Positron emission tomography: current challenges and opportunities for technological advances in clinical and preclinical imaging systems. Annu. Rev. Biomed. Eng. 2015, 17, 385-414.
  4. Debnath, S.; Hao, G.; Guan, B.; Thapa, P.; Hao, J.; Hammers, H; Sun, X. Theranostic small-molecule prodrug conjugates for targeted delivery and controlled release of toll-like receptor 7 agonists. Int. J. Mol. Sci. 2022, 23, 7160.

(Q6) For the comment, “(6) In line 58, please include the references;

https://doi.org/10.1053/j.semnuclmed.2017.05.006

https://doi.org/10.3390/ph14060530

https://doi.org/10.1158/1538-7445.AM2022-2478.”,

(A6) Several references was inserted and cited.

  1. Donnelly, D.J. Small molecule PET tracers in drug discovery. Semin. Nucl. Med. 2017, 47, 454-460.
  2. Ozenil, M.; Aronow, J.; Millard, M.; Langer, T; Wadsak, W; Hacker, M; Pichler, V. Update on PET tracer development for muscarinic acetylcholine receptors. Pharmaceuticals 2021, 14, 530.

27.Debnath, S.; Stevens, C.; Brandenburg, O.; Sovich, J.; Gonzalez, P.; Qin, Q.; Haldeman, S.; Tcheuyap, V.T.; Christie, A.; Thapa, P.; et al. Development of a novel HIF2a PET tracer: From proof of concept to a clinical trial. Cancer Res. 2022, 82, 2478.

(Q7) For the comment, “(7) In line 69, please include the reference from

Bioconjug Chem. 2015 Jan 21; 26(1): 1–18. doi: 10.1021/bc500475e

Int. J. Mol. Sci. 2022, 23(3), 1158; https://doi.org/10.3390/ijms23031158”,

(A7) Several references was inserted and cited.

  1. Jacobson, O.; Kiesewetter, D.O.; Chen, X. Fluorine-18 radiochemistry, labeling strategies and synthetic routes. Bioconjug. Chem. 2015, 26, 1-8.
  2. Debnath, S.; Zhou, N.; McLaughlin, M.; Rice, S.; Pillai, A.K.; Hao, G.; Sun, X. PSMA-Targeting imaging and theranostic agents-current status and future perspective. Int. J. Mol. Sci. 2022, 23, 1158.

(Q8) For the comment, “(8) In line 110, please elaborate the effect of temperature and species on binding affinity of Ro5-4864.”,

(A8) We checked literatures, and found the following sentence from the literatures

Binding of Ro5-4864 is temperature sensitive with maximum binding at 0°C (Wang JK, Taniguchi T, Spector S. Properties of [3H]diazepam binding sites on rat blood platelets. Life Sci. 1980;27:1881–88)

 Ro5-4864 could bind to human and mouse PBR, but not to bovine PBR (Farges R, Joseph Liauzun E, Shire D, Caput D, Le Fur G, Ferrara P. Site-directed mutagenesis of the peripheral benzodiazepine receptor: identification of amino acids implicated in the binding site of Ro5-4864. Life Sci. 1980;27:1881–88))

Therefore, these references were inserted and cited.

  1. Wang, J.K.; Taniguchi, T.; Spector. S. Properties of [3H]diazepam binding sites on rat blood platelets. Life Sci. 1980, 27, 1881–1888
  2. Farges, R.; Joseph Liauzun, E.; Shire, D.; Caput, D.; Le Fur, G.; Ferrara, P. Site-directed mutagenesis of the peripheral benzo-diazepine receptor: identification of amino acids implicated in the binding site of Ro5-4864. Mol. Pharmacol. 1994, 46, 1160-1167

(Q9) For the comment, “(9) In line 130, please mention the baseline SUV value at 5 min.”,

(A9) We checked literatures to find the baseline SUV value at 5 min, however, the references do not have baseline SUV value at 5 min.

(Q10) For the comment, “(10) It would be helpful to know the value of ‘n’ for radiochemical yield and molar activity mentioned throughout the report.”,

(A10) We checked information in references and added the value of ‘n’ for radiochemical yield and molar activity throughout the report:

(Q11) For the comment, “(11) It does not explained the “higher in vivo stability” in line 190, target-to-background ratio is not supporting the stability.”

(A11)

The sentences “This result indicates higher in vivo stability for [18F]fluoromethyl-PBR28-d2 compared with [18F]fluoromethyl-PBR28” was removed. Therefore, the sentences “However, uptake of [18F]fluoromethyl-PBR28-d2 was reduced in the skull and femur compared with [18F]fluoromethyl-PBR28 (1.5% ± 1.2% versus 4.1% ± 1.7% of injected dose (ID)/g at 2 h post-injection), and cleared more rapidly in the contralateral area, indicating an improved target-to-background ratio (approximately 3.8-fold for [18F]fluoromethyl-PBR28-d2 versus 3.0-fold for [18F]fluoromethyl-PBR28 at 30 min post-injection). This result indicates higher in vivo stability for [18F]fluoromethyl-PBR28-d2 compared with [18F]fluoromethyl-PBR28.” was changed to “However, uptake of [18F]fluoromethyl-PBR28-d2 was reduced in the skull and femur compared with [18F]fluoromethyl-PBR28 (1.5% ± 1.2% versus 4.1% ± 1.7% of injected dose (ID)/g at 2 h post-injection), and cleared more rapidly in the contralateral area, indicating an improved target-to-background ratio (approximately 3.8-fold for [18F]fluoromethyl-PBR28-d2 versus 3.0-fold for [18F]fluoromethyl-PBR28 at 30 min post-injection).”

(Q12) For the comment, “(12) Please mention the metabolism information in line 195.”,

(A12) The metabolism information was mentioned in next sentences in line 195, 196. For clearly, The sentences “In 2015, N-{2-[2-18F-Fluoroethoxy]-5-metoxybenzyl}-N-[2-(4 metoxyphenoxy) pyri-din-3-yl]axetamit ([18F]FEMPA) (5) was used as a PET tracer of TSPO in AD patients [32], [18F]FEMPA undergoing rapid metabolism. At 20 min post-injection, only 20% of the tracer was detectable in the plasma, and this number decreased to 10% at 90 min” were changed to ““In 2015, N-{2-[2-18F-Fluoroethoxy]-5-metoxybenzyl}-N-[2-(4 metoxyphenoxy) py-ri-din-3-yl]axetamit ([18F]FEMPA) (5) was used as a PET tracer of TSPO in AD patients [53]. [18F]FEMPA undergone rapid metabolism at 20 min post-injection, and more than 20% of the tracer was detectable in the plasma, and this number decreased to 10% at 90 min"

(Q13) For the comment, “(13) The statement in line 226 to 228 is not suitable. Purity at 25 °C for 4 h, does not confirm the tracer is suitable for in vivo/in vitro evaluation.”,

(A13) The sentences “which means that [18F]FEDAA1106 was stable enough for evaluation [35]” was removed. Therefore, the sentences “After maintaining it at 25 °C for 4 h, the radiochemical purities of [18F]FEDAA1106 were still >95%, which means that [18F]FEDAA1106 was stable enough for evaluation [35].” was changed to “After maintaining it at 25 °C for 4 h, the radiochemical purities of [18F]FEDAA1106 were still >95%  [35].”.

(Q14) For the comment, “(14) Please crosscheck the molar activity value in line 270.”

(A14) We checked and correct them, and the following sentences “The molar activity of [18F]DAA1106 was 60–100 GBq/mol” was changed to “The molar activity of [18F]DAA1106 was 60–100 GBq/μmol”.

(Q15) For the comment, “(15) In line 271, radiochemical 271 purity at room temperature for 120 min does not confirm the tracer is good for clinical use.”

(A15) The sentences “suggesting that it is appropriate for clinical use” was removed. Therefore, the sentences “After being kept at room temperature for 120 min, the radiochemical purity of [18F]DAA1106 remained above 95%, suggesting that it is appropriate for clinical use [43]” was changed to “After being kept at room temperature for 120 min, the radiochemical purity of [18F]DAA1106 remained above 95% [43]”.

(Q16) For the comment, “(16) In line 289, please change “measured radioactivity” to “molar activity”.”,

(A16) The words “measured radioactivity” was changed to “molar activity”

(Q17) For the comment, “(17) In line 383, Please correct “lipophilicity (log P = 7.5)”.”,

(A17) We checked and correct them. The words “lipophilicity (log P = 7.5)” was changed to “lipophilicity (at pH = 7.5)”

(Q18) For the comment, “(18) Please mention the IC50 value in line 397.”,

(A18) The IC50 value was added as your recommendation. So that, the sentence: “Both VUIIS1009A and VUIIS1009B have high TSPO binding affinity, with measured IC50  values 1/500th that of DPA-714” was changed to:” Both VUIIS1009A and VUIIS1009B have high TSPO binding affinity, with measured IC50  values 1/500th that of DPA-714 (IC50 = 14.4 pM for VUIIS1009A and IC50 = 19.4 pM for VUIIS1009B)”.

(Q19) For the comment, “(19) Please mention the tumors category in line 763.”,

(A19) The tumors category was mentioned as your recommendation. So that, a sentence:” A PET study demonstrated that [18F]AB5186 was clearly binding to tumors, and its location was consistent with that reported by histology.” was changed to :“A PET study on the mouse bearing an intracranial U87MG-Luc2 tumor demonstrated that [18F]AB5186 was clearly binding to tumors, and its location was consistent with that reported by histology”.

Reviewer 3 Report

The manuscript (pharmaceutics-1941974) attempted to summary the history and advances in the development of 18F-radiolabeled PET tracers targeting TSPO. It is useful to have an update on the status of F18-labelled TSPO tracers.

Overall, the reviewer feel that the review article is poorly written. I have the following more specific comments:

1.      It would be helpful to have a table summarizing the TSPO tracers and their key properties so that the readers can have a quick look on their pros and cons.

2.      It’s less helpful to just list the tspo tracers. Can the authors give their opinion and assessment on these ligands to help the readers understand the literature and the status of TSPO ligands and imaging?

3.      For a review of TSPO PET radiotracer, it should be balanced on the advantage and disadvantage of TSPO as a biomarker of neuroinflammation or microglial activation. It is now with consensus that TSPO is not specific for microglia, in particular in chronic human conditions, and cannot discriminate different glial states including M1 vs M2 microglia.

4.      There are many inaccurate citation of the literature. For example, references #3 and #7, references #10 and #24 are duplicated; reference #29 used first names, instead of last names, in the author list – which reference management SW is used by the authors? In citing reference #26 (page4, line 141), the percentage of FEPPA metabolites in brain at 120 min was 23% instead of 27%. There may be other such errors that the reviewer missed; but a review article has to be accurate in citing the literature so as not to be misleading.

5.      What is advantage of F18 over C11 with a lower positron energy (650 keV), page 2, line 70? Is there a reference?

6.      Tspo is actually an abundant protein in human brain, although its true function is still debated (see Tong et al. 2020. Concentration, distribution, and influence of aging on the 18 kDa translocator protein in human brain: Implications for brain imaging studies. J Cereb Blood Flow Metab. 2020 May;40(5):1061-1076; Lee et al. 2020. Translocator protein (TSPO): the new story of the old protein in neuroinflammation. BMB Rep. 2020 Jan;53(1):20-27; Selvaraj and Stocco 2015 The changing landscape in translocator protein (TSPO) function. Trends Endocrinol Metab. 2015 Jul;26(7):341-8).

7.      Page 2, line 87-92: “resting state” instead of “basic state”; “A high concentration of microglial cells in the CNS, particularly at inflammatory sites, is considered a biomarker of neuroinflammation” is not accurate – morphology change of microglia is actually more critical here; “Because high TSPO expression is closely related to the activation of microglia [19]” – activated microglia do not necessarily over-express TSPO (see Nutma et al. 2021 Activated microglia do not increase 18 kDa translocator protein (TSPO) expression in the multiple sclerosis brain. Glia. 2021 Oct;69(10):2447-2458).

Author Response

Reviewer #3’s comments

(Q1) For the comment, “1. It would be helpful to have a table summarizing the TSPO tracers and their key properties so that the readers can have a quick look on their pros and cons.”,

(A1) We added a table summarizing the TSPO tracers and their key properties (Table 1. 18F-Radiolabeled TSPO ligands) into Page 04.

(Q2) For the comment, “2. It’s less helpful to just list the tspo tracers. Can the authors give their opinion and assessment on these ligands to help the readers understand the literature and the status of TSPO ligands and imaging?”,

(A2) We added comments in Table 1 to provide our opinion and assessment on each TSPO ligands.

(Q3) For the comment, “3. For a review of TSPO PET radiotracer, it should be balanced on the advantage and disadvantage of TSPO as a biomarker of neuroinflammation or microglial activation. It is now with consensus that TSPO is not specific for microglia, in particular in chronic human conditions, and cannot discriminate different glial states including M1 vs M2 microglia.”,

(A3) We appreciate the reviewer's comments, and the following sentences have been added into “conclusion” part (Page 127) to describe the disadvantage of TSPO as a biomarker of neuroinflammation or microglial activation: “One of the TSPO-PET imaging method's alleged drawbacks is that there are still few clinical studies and in-depth investigations into the cell types accountable for the TSPO response in several brain diseases. The research about TSPO on human are often extrapolated from rodent studies. However, many recent researches suggested that the expression of TSPO in animal models (preclinical investigations) may not be the same as the outcomes obtained in human.[34,139]. The sensitivity of most second-generation radioligands to the TSPO polymorphism (rs6971), which changes the TSPO affinity for the tracers, is other potential limitation of TSPO-PET neuroimaging investigations. Moreover, several recent studies have demonstrated that the expression of TSPO is not only  represents for microglia or inflammatory response,  but also involved in many other biological activities in the cell such as cellular metabolism, energy homeostasis, or oxidative stress during inflammation [140]”

Several references was inserted and cited.

  1. Nutma, E.; Gebro, E.; Marzin, M.C.; van der Valk, P.; Matthews, P.M.; Owen, D.R.; Amor S. Activated microglia do not increase 18 kDa translocator protein (TSPO) expression in the multiple sclerosis brain. Glia 2021, 69, 2447-2458
  • Owen, D.R.; Narayan, N.; Wells, L.; Healy, L.; Smyth, E.; Rabiner, E.A.; Galloway, D.; Williams, J.B.; Lehr, J.; Mandhair, H.; et al. Pro-inflammatory activation of primary microglia and macrophages increases 18 kDa translocator protein expression in rodents but not humans. Cereb. Blood Flow Metab. 2017, 37, 2679-2690
  • Notter, T.; Coughlin, J.M.; Sawa, A.; Meyer, U. Reconceptualization of translocator protein as a biomarker of neuroinflammation in psychiatry.  Psychiatry 2018, 23, 36-47

(Q4) For the comment, “4. There are many inaccurate citation of the literature. For example, references #3 and #7, references #10 and #24 are duplicated; reference #29 used first names, instead of last names, in the author list – which reference management SW is used by the authors? In citing reference #26 (page4, line 141), the percentage of FEPPA metabolites in brain at 120 min was 23% instead of 27%. There may be other such errors that the reviewer missed; but a review article has to be accurate in citing the literature so as not to be misleading.”

(A4-1) The citations of the literatures were corrected in a uniform format as your recommendation

(A4-2) The word “27%” was changed to “23%”. (Due to manuscript editing, its location changed from page 4 to page 6).

(Q5) For the comment, “5. What is advantage of F18 over C11 with a lower positron energy (650 keV), page 2, line 70? Is there a reference?”,

(A5) The lower positron energy of 18F results in a short diffusion range (<2.4 mm) that favorably increases the resolution limits of the PET images. (while 11C is 1.0 MeV)

Several references was inserted and cited.

  1. Pretze, M.; Große-Gehling, P.; Mamat, C. Cross-coupling reactions as valuable tool for the preparation of PET radiotracers. Molecules 2011, 16, 1129-1165.
  2. Jacobson, O.; Kiesewetter, D.O.; Chen, X. Fluorine-18 radiochemistry, labeling strategies and synthetic routes. Bioconjug. Chem. 2015, 26, 1-8.
  3. Li, M.; Zelchan, R.; Orlova, A. The Performance of FDA-Approved PET Imaging Agents in the Detection of Prostate Cancer. Biomedicines 2022, 10, 2533.

(Q6) For the comment, “6. Tspo is actually an abundant protein in human brain, although its true function is still debated (see Tong et al. 2020. Concentration, distribution, and influence of aging on the 18 kDa translocator protein in human brain: Implications for brain imaging studies. J Cereb Blood Flow Metab. 2020 May;40(5):1061-1076; Lee et al. 2020. Translocator protein (TSPO): the new story of the old protein in neuroinflammation. BMB Rep. 2020 Jan;53(1):20-27; Selvaraj and Stocco 2015 The changing landscape in translocator protein (TSPO) function. Trends Endocrinol Metab. 2015 Jul;26(7):341-8).”,

(A6) Several sentences have been added on “introduction” part (page 02) to further clarify the function of TSPO in the human brain: “However, recent studies evaluating TSPO function through genetics have raised questions about the true role of TSPO in steroidogenesis, as well as several other functions [9-11]. So, these TSPO functions need to be carefully re-evaluated.”

Several references was inserted and cited:

  1. Tong, J.; Williams, B.; Rusjan, P.M.; Mizrahi, R.; Lacapère, J.J.; McCluskey, T.; Furukawa, Y.; Guttman, M.; Ang, L.C.; Boileau, I.; et al. Concentration, distribution, and influence of aging on the 18 kDa translocator protein in human brain: implications for brain imaging studies. Cereb. Blood Flow Metab. 2020, 40, 1061-1076.
  2. Lee, Y.; Park, Y.; Nam, H.; Lee, J.W.; Yu, S.W. Translocator protein (TSPO): the new story of the old protein in neuroinflammation. BMB Rep. 2020, 53, 20.
  3. Selvaraj, V.; Stocco, D.M. The changing landscape in translocator protein (TSPO) function. Trends Endocrinol. Metab. 2015, 26, 341-348.

(Q7) For the comment, “7. Page 2, line 87-92: “resting state” instead of “basic state”; “A high concentration of microglial cells in the CNS, particularly at inflammatory sites, is considered a biomarker of neuroinflammation” is not accurate – morphology change of microglia is actually more critical here; “Because high TSPO expression is closely related to the activation of microglia [19]” – activated microglia do not necessarily over-express TSPO (see Nutma et al. 2021 Activated microglia do not increase 18 kDa translocator protein (TSPO) expression in the multiple sclerosis brain. Glia. 2021 Oct;69(10):2447-2458).”

(A7)-1 At page 2, the word “basic state” was replaced by “resting state”.

(A7)-2 Some sentences have been modified and added to clarify arguments about the relationship between the TSPO expression and activation of microglia.

The sentences: “A high concentration of microglial cells in the CNS, particularly at inflammatory sites, is considered a biomarker of neuroinflammation” was changed to “The activation of microglial cells in the CNS, particularly at inflammatory sites, is considered as a biomarker of neuroinflammation”

The sentences :” Because high TSPO expression is closely related to the activation of microglia [19], and several studies have found that increased expression of TSPO represent activation of microglial cells or increase neuroinflammation [20], TSPO is considered a useful biomarker of neuroinflammation and related diseases“ were changed to :”Although previous studies showed that the high TSPO expression is closely related to the activation of microglia [36], and many studies suggested that increased expression of TSPO represent activation of microglial cells or increased neuroinflammation [34], but a few recent studies showed that high TSPO expression do not necessarily cause over expression of microglia activation [35]. However, TSPO is still considered a useful biomarker of neuroinflammation and related diseases”

Several references was inserted and cited.

  1. Nutma, E.; Gebro, E.; Marzin, M.C.; van der Valk, P.; Matthews, P.M.; Owen, D.R.; Amor S. Activated microglia do not increase 18 kDa translocator protein (TSPO) expression in the multiple sclerosis brain. Glia 2021, 69, 2447-2458.

Round 2

Reviewer 3 Report

The reviewer thank the authors for their careful revision. I have only a few minor comments listed as follows, mainly on literature citing so as to be consistent and fair to the literature:

1.      Page 3, line 98-99: “a few recent studies showed that high TSPO expression do not necessarily cause over expression of microglia activation [35]” should be “a few recent studies showed that microglia activation is not necessarily associated with over-expression of TSPO in individual microglial cells [35].”

2.      Page 4, Table 1, PBR06: comment says “Effective in preclinical and clinic studies” but then “Lack of clinical studies”. Actually PBR06 has been used in clinical studies of MS patients by Singhal et al. 2018-2020;

3.      Page 4, Table 1, DPA-714: “Lavisse et al. Increased microglial activation in patients with Parkinson disease using [18F]-DPA714 TSPO PET imaging. Parkinsonism Relat Disord 2021;82:29-36” examined PD patients; “Van Weehaeghe et al. J Nucl Med. Moving Toward Multicenter Therapeutic Trials in Amyotrophic Lateral Sclerosis: Feasibility of Data Pooling Using Different Translocator Protein PET Radioligands. 2020;61(11):1621-1627” examined ALS patients; “Lee et al. Translocator Protein (18 kDa) Polymorphism (rs6971) in the Korean Population. Dement Neurocogn Disord. 2022;21(2):71-78”, “Hamelin et al. Distinct dynamic profiles of microglial activation are associated with progression of Alzheimer's disease. Brain. 2018;141(6):1855-1870”, and “Hamelin et al. Early and protective microglial activation in Alzheimer's disease: a prospective study using 18F-DPA-714 PET imaging. Brain 2016;139(Pt 4):1252-64” also examined AD patients and showed increased TSPO binding, NOT “ineffective” in this condition;

4.      PBR111 has been employed in many clinical studies of MS and schizophrenia.

5.      GE180 was showed by Zanotti-Fregonara et al. (Head-to-Head Comparison of 11C-PBR28 and 18F-GE180 for Quantification of the Translocator Protein in the Human Brain. J Nucl Med. 2018;59(8):1260-1266) to be unfavorable for brain imaging due to poor brain penetration;

6.      Page 28, line 863: “…is other potential limitation…” – change ‘other’ to ‘another’.

Author Response

(Q1) For the comment, “1. Page 3, line 98-99: “a few recent studies showed that high TSPO expression do not necessarily cause over expression of microglia activation [35]” should be “a few recent studies showed that microglia activation is not necessarily associated with over-expression of TSPO in individual microglial cells [35].”

(A1) The sentence “a few recent studies showed that high TSPO expression do not necessarily cause over expression of microglia activation [35]” was changed to “a few recent studies showed that microglia activation is not necessarily associated with over-expression of TSPO in individual microglial cells [35]”.

(Q2) For the comment, “2. Page 4, Table 1, PBR06: comment says “Effective in preclinical and clinic studies” but then “Lack of clinical studies”. Actually PBR06 has been used in clinical studies of MS patients by Singhal et al. 2018-2020.”

(A2) The works “Lack of clinical studies” were removed.

Several references were inserted and cited

  1. Singhal, T.; O'Connor, K.; Dubey, S.; Pan, H.; Chu, R.; Hurwitz, S.; Cicero, S.; Tauhid, S.; Silbersweig, D.; Stern, E.; et al. Gray matter microglial activation in relapsing vs progressive MS: A [F-18] PBR06-PET study. Neurol.: Neuroimmunol. NeuroInflammation. 2019, 6, 5.

  1. Singhal, T.; Cicero, S.; Pan, H.; Carter, K.; Dubey, S.; Chu, R.; Glanz, B.; Hurwitz, S.; Tauhid, S.; Park, M.A.; et al. Regional microglial activation in the substantia nigra is linked with fatigue in MS. Neurol.: Neuroimmunol. NeuroInflammation. 2020, 7, 5.

(Q3) For the comment, “3. Page 4, Table 1, DPA-714: “Lavisse et al. Increased microglial activation in patients with Parkinson disease using [18F]-DPA714 TSPO PET imaging. Parkinsonism Relat Disord 2021;82:29-36” examined PD patients; “Van Weehaeghe et al. J Nucl Med. Moving Toward Multicenter Therapeutic Trials in Amyotrophic Lateral Sclerosis: Feasibility of Data Pooling Using Different Translocator Protein PET Radioligands. 2020;61(11):1621-1627” examined ALS patients; “Lee et al. Translocator Protein (18 kDa) Polymorphism (rs6971) in the Korean Population. Dement Neurocogn Disord. 2022;21(2):71-78”, “Hamelin et al. Distinct dynamic profiles of microglial activation are associated with progression of Alzheimer's disease. Brain. 2018;141(6):1855-1870”, and “Hamelin et al. Early and protective microglial activation in Alzheimer's disease: a prospective study using 18F-DPA-714 PET imaging. Brain 2016;139(Pt 4):1252-64” also examined AD patients and showed increased TSPO binding, NOT “ineffective” in this condition.”,

(A3) The words “PD patients, ALS Patients” were added to Table 1 of Page 4.

The sentence “Effective in monitoring and diagnosis for PACNS, stroke and MS patients, but ineffective in AD patients” was changed to “Effective in monitoring and diagnosis for many neurological diseases”.

Several references were inserted and cited

  1. Lavisse, S.; Goutal, S.; Wimberley, C.; Tonietto, M.; Bottlaender, M.; Gervais, P.; Kuhnast, B.; Peyronneau, M.A.; Barret, O.; Lagarde, J.; et al. Increased microglial activation in patients with Parkinson disease using [18F]-DPA714 TSPO PET imaging. Park. Relat. Disord 2021, 82, 29-36.

  1. Van Weehaeghe, D.; Babu. S.; De Vocht, J.; Zürcher, N.R.; Chew, S.; Tseng, C.E.; Loggia, M.L.; Koole, M.; Rezaei, A.; Schramm, G.; et al. Moving toward multicenter therapeutic trials in amyotrophic lateral sclerosis: feasibility of data pooling using different translocator protein PET radioligands. J. Nucl. Med. 2020, 61, 1621-1627.

  1. Lee, H.; Noh, Y.; Kim. W.R.; Seo, H.E.; Park, H.M. Translocator Protein (18 kDa) Polymorphism (rs6971) in the Korean Population. Dement. Neurocogn. Disord. 2022, 21, 71.

  1. Hamelin, L.; Lagarde, J.; Dorothée, G.; Leroy, C.; Labit, M.; Comley, R.A.; de Souza, L.C.; Corne, H.; Dauphinot, L.; Bertoux, M.; et al. Early and protective microglial activation in Alzheimer’s disease: a prospective study using 18F-DPA-714 PET imaging. Brain 2016, 139, 1252-1264.

  1. Hamelin, L.; Lagarde, J.; Dorothée, G.; Potier, M.C.; Corlier, F.; Kuhnast, B.; Caillé, F.; Dubois, B.; Fillon, L.; Chupin, M.; et al. Distinct dynamic profiles of microglial activation are associated with progression of Alzheimer's disease. Brain 2018, 141, 1855-1870.

(Q4) For the comment: “4. PBR111 has been employed in many clinical studies of MS and schizophrenia.”

(A4) The words “Schizophrenia patients, Psychosis patients, MS patients” were added to Table 1 of Page 4.

The works “Effective in clinical application” were added.

Several references were inserted and cited:

  1. Ottoy, J.; De Picker, L.; Verhaeghe, J.; Deleye, S.; Kosten, L.; Sabbe, B.; Coppens, V.; Timmers, M.; Van Nueten, L.; Ceyssens, S.; et al. 18F-PBR111 PET imaging in healthy controls and schizophrenia: test–retest reproducibility and quantification of neuroinflammation. J. Nucl. Med. 2018, 59, 1267-1274.

  1. De Picker, L.; Ottoy, J.; Verhaeghe, J.; Deleye, S.; Fransen, E.; Kosten, L.; Sabbe, B.; Coppens, V.; Timmers, M.; de Boer, P.; et al. State-associated changes in longitudinal [18F]-PBR111 TSPO PET imaging of psychosis patients: evidence for the accelerated ageing hypothesis?. Brain Behav. Immun. 2019, 77, 46-54.

  1. Datta, G.; Colasanti, A.; Kalk, N.; Owen, D.; Scott, G; Rabiner, E.A.; Gunn, R.N.; Lingford-Hughes, A.; Malik, O.; Ciccarelli, O.; Nicholas, R. 11C-PBR28 and 18F-PBR111 detect white matter inflammatory heterogeneity in multiple sclerosis. J. Nucl. Med. 2017, 58, 1477-1482.

(Q5) For the comment, “5. GE180 was showed by Zanotti-Fregonara et al. (Head-to-Head Comparison of 11C-PBR28 and 18F-GE180 for Quantification of the Translocator Protein in the Human Brain. J Nucl Med. 2018;59(8):1260-1266) to be unfavorable for brain imaging due to poor brain penetration.”

(A5) The words “poor brain penetration in clinical study” were added to Table 1 of Page 4.

A reference was inserted and cited.

  1. Zanotti-Fregonara, P.; Pascual, B.; Rizzo, G.; Yu, M.; Pal, N.; Beers. D.; Carter, R.; Appel, S.H.; Atassi, N.; Masdeu, J.C. Head-to-head comparison of 11C-PBR28 and 18F-GE180 for quantification of the translocator protein in the human brain. J. Nucl. Med. 2018, 59, 1260-1266.

(Q6) For the comment, “6. Page 28, line 863: “…is other potential limitation…” – change ‘other’ to ‘another’.”,

(A6) The words “is other potential limitation” was changed to “is another potential limitation”.
